# DeFT: Decoding with Flash Tree-Attention for Efficient Tree-structured LLM Inference

**Jinwei Yao**[1,4,*] **Kaiqi Chen**[2,*] **Kexun Zhang**[3] **Jiaxuan You** [4,†]
**Binhang Yuan**[5] **Zeke Wang**[2,†] **Tao Lin**[1,†]
jinwei.yao1114@gmail.com; {chiaki_cage,wangzeke}@zju.edu.cn;
kexunz@andrew.cmu.edu; jiaxuan@illinois.edu;
biyuan@ust.hk; lintao@westlake.edu.cn
[1]Westlake University [2]Zhejiang University [3]Carnegie Mellon University
[4]University of Illinois Urbana-Champaign [5]Hong Kong University of Science and Technology

## Abstract

Large language models (LLMs) are increasingly employed for complex tasks that process multiple generation calls in a tree structure with shared prefixes of tokens, including few-shot prompting, multi-step reasoning, speculative decoding, etc. However, existing inference systems for tree-based applications are inefficient due to improper partitioning of queries and KV cache during attention calculation. This leads to two main issues: (1) a lack of memory access (IO) reuse for KV cache of shared prefixes, and (2) poor load balancing. As a result, there is redundant KV cache IO between GPU global memory and shared memory, along with low GPU utilization. To address these challenges, we propose DeFT[1] (Decoding with Flash Tree-Attention), a hardware-efficient attention algorithm with prefix-aware and load-balanced KV cache partitions. DeFT reduces the number of read/write operations of KV cache during attention calculation through *KV-Guided Grouping*, a method that avoids repeatedly loading KV cache of shared prefixes in attention computation. Additionally, we propose *Flattened Tree KV Splitting*, a mechanism that ensures even distribution of the KV cache across partitions with little computation redundancy, enhancing GPU utilization during attention computations. By reducing 73-99% KV cache IO and nearly 100% IO for partial results during attention calculation, DeFT achieves up to 2.23/3.59× speedup in the decoding/attention latency across three practical tree-based workloads compared to state-of-the-art attention algorithms. Our code is available at https://github.com/LINs-lab/DeFT.

## 1 Introduction

Large language models (LLMs) (Achiam et al., 2023; Touvron et al., 2023a;b) are extensively utilized across a range of tasks like chatbot (Roller et al., 2020), code generation (Mark et al., 2021), reasoning (Yao et al., 2023; Besta et al., 2023; Ning et al., 2023), etc. Traditionally, the interactions between LLMs and application users are sequential: the user sends a new prompt after completion result of the previous prompt is received. However, many applications are now designed to process sequences with an internal tree structure, including self-consistency (Wang et al., 2022), few-shot prompting (Mann et al., 2020), multi-step reasoning (Yao et al., 2023; Hao et al., 2023; Xie et al., 2024), and speculative decoding (Miao et al., 2023; Cai et al., 2024), etc, as shown in Figure 1. Usually, these applications produce substantially more tokens than traditional ones, to provide large space for tree search (Graves, 2012; Lu et al., 2022; Liu et al., 2023) or selection, as shown in Table 1. **We need a more efficient decoding algorithm in response to this interaction paradigm change from sequence-based decoding to tree-based decoding**.

---

[*]Equal contribution. Work done during Jinwei's visit to Westlake University.

[†]Corresponding author.

[1]By default, DeFT refs to DeFT-Flatten, which has **Flattened Tree KV Splitting** before loading KV cache for attention calculation.

When requests have shared prefixes in a tree structure, existing inference systems (Hugging Face; NVIDIA; Kwon et al., 2023) designed for sequence-based decoding introduce redundancy by failing to be prefix-aware at one or more of the following three levels: (1) *computation*—for instance, the redundant recomputation of KV caches for shared prompts across requests in a batch (Hugging Face); (2) *memory storage*—for example, the redundant storage of KV caches for shared prefixes (Hugging Face; Kwon et al., 2023; NVIDIA); (3) *memory access (IO)*—such as repeatedly loading the KV cache of a shared system prompt during attention calculations (Hugging Face; Kwon et al., 2023; NVIDIA). Although some tree-based inference systems (Zheng et al., 2023; Gim et al., 2023; Cai et al., 2024; Miao et al., 2023) address the first two issues, they largely overlook the third and arguably the most crucial aspect: *memory access*, which is critical in the context of memory-bound LLM inference (Shazeer, 2019; Cai et al., 2024; Kim et al., 2023).

Table 1: **Comparison of efficiency in sequence-based CoT (Wei et al., 2022) and tree-based ToT (Yao et al., 2023) decoding for a reasoning task.** The task is *sorting* 128 *numbers* from Besta et al. (2023). The total generated tokens of CoT is only 525 while 38,315 in ToT, resulting in inefficiency in end-to-end latency (`second`) and IO (`TB`). IO mainly consists of two parts as follows. (i) *KV cache*: `IO-KV`; (ii) *Partial results during attention calculation like* $QK^T$ *and softmax*: `IO-PA`; Baselines: (i) *Flash-Decoding* (Dao et al., 2023); (ii) *Tree Attention*: tree attention in Medusa (Cai et al., 2024).

| | Latency | IO-KV | IO-PA |
|---|---|---|---|
| Flash-Decoding + CoT | 21 | 0.6 | 0 |
| Flash-Decoding + ToT | 429.65 | 59.96 | 0 |
| Tree Attention + ToT | 380.87 | 12.40 | 3.69 |
| DeFT-Flatten(ours) + ToT | 94.67 | 12.40 | 0 |
| Speed up over best baseline | 4.02× | - | - |

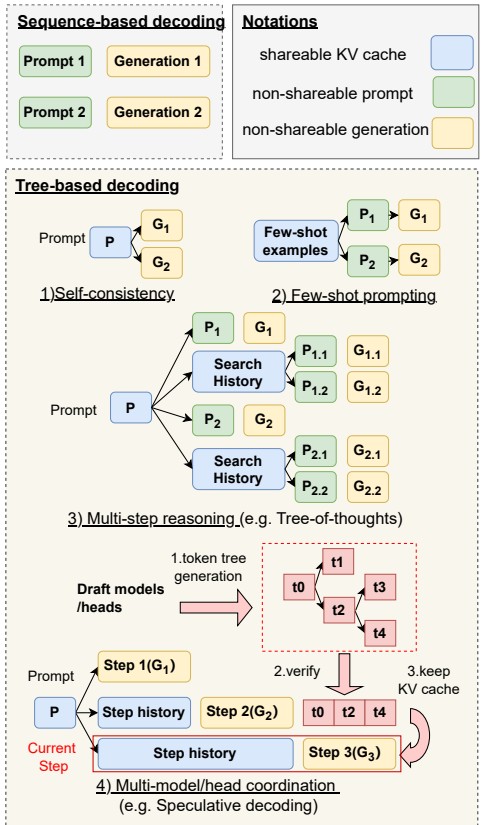

Figure 1: **An illustration of Sequence-based decoding and Tree-based decoding**.

To accelerate the tree-structured LLM inference, an important question is whether we can leverage the shared patterns in multi-cascaded prefixes to design a faster and more memory-efficient attention algorithm. This task is challenging due to two key issues as follows. **C1: How to ensure prefix-awareness in memory access of KV cache?** Current memory-efficient attention algorithms (Dao et al., 2022; 2023; Hong et al., 2023) are optimized for sequence-based decoding, which leads to a lack of prefix-awareness during memory access. As a result, shared prefixes in the KV cache are repeatedly loaded. **C2: How to split the tree-structured KV cache for load balancing and high GPU utilization?** For optimal GPU utilization, the current KV splitting strategy for sequence-based decoding—Flash-Decoding (Dao et al., 2023), which splits sequence KV into chunks—cannot be directly applied to tree-structured KV. Tree-structured KV caches also need to be effectively partitioned: however, if we naively split them by nodes, token lengths across different nodes can vary significantly (e.g., in speculative decoding (Cai et al., 2024), some nodes might only have 1 token while the root node could have thousands), making it difficult to maintain load balance and efficient computation.

To address the above challenges, we propose DEFT-Flatten, a prefix-aware tree attention algorithm with a flattened tree KV splitting strategy, based on two key insights. ● First, how queries and KV caches are grouped for attention calculation significantly impacts memory access. Existing approaches use a ***Q-Guided Grouping*** strategy, where each request/query is grouped with all corresponding KV caches. While this eliminates IO redundancy for queries, the prefix KV cache still gets loaded multiple times. To address **C1**, we propose ***KV-Guided Grouping***: DEFT-Flatten groups the prefix's KV cache with all shared queries, ensuring the prefix KV cache is only loaded once, significantly reducing redundant loading with negligible IO overhead for reloading queries. The IO overhead for queries (Q) is minimal compared to the KV cache, as the maximum query length

typically corresponds to the number of root-to-leaf paths in the tree, making the queries relatively short (e.g., dozens of tokens) compared to the KV cache length in each node (e.g., hundreds or thousands of tokens). ● Second, since LLM inference is IO-bound, the attention overhead of each QKV group is primarily influenced by the IO of the KV cache. Therefore, it is crucial to ensure that the KV lengths of different QKV groups are nearly balanced. To address **C2**, we propose a ***Flattened Tree KV Splitting***, which enables balanced partitions by dividing the flattened tree KV into even chunks, using bit causal masks to capture causal relationships between queries and KV cache.

We summarize our contributions as follows:

- We propose a hardware-efficient tree attention algorithm—DEFT-Flatten, which is IO-aware of shared prefixes' KV cache and load-balanced in computation.
- We implement DEFT-Flatten on OpenAI Triton (Tillet et al., 2019) to gain precise management over memory access and fuse all attention operations into a single GPU kernel.
- We theoretically justify the superiority of DEFT-Flatten over the existing attention algorithms (Wolf et al., 2019; Dao et al., 2023; Cai et al., 2024; Miao et al., 2023) in terms of IO complexity.
- We empirically verify its effectiveness on few-shot prompting, multi-step reasoning, and speculative-decoding tasks. DEFT-Flatten can achieve a decoding latency speedup of **1.3×** for few-shot prompting, **2.2×** for speculative decoding, **1.1×** for multi-step reasoning, due to an up to **3.59×** faster attention calculation, with the baseline implementations (Dao et al., 2023; Cai et al., 2024; Zheng et al., 2023).
- We compare different tree split strategies—DEFT-Node, DEFT-Node-Chunk, and DEFT-Flatten in ablation studies (see section 4.4), showing the balanced partitioning of QKV groups matters.

## 2 RELATED WORK

**Tree-based Decoding.** Tree-based decoding, exemplified by beam search (Graves, 2012), has been pivotal in NLP, handling lexical and logical constraints (Anderson et al., 2017; Post & Vilar, 2018; Hokamp & Liu, 2017), mitigating gender bias (Lu et al., 2021), achieving communicative goals (Holtzman et al., 2018), and improving alignment (Liu et al., 2023). Based on the structure feature of queries and KV cache, we can classify tree-based decoding into two patterns: (i) Tree-structured past KV with parallel queries—usually in multi-step reasoning (Yao et al., 2023; Besta et al., 2023; Ning et al., 2023), using search trees with parallel hypothesis generation and selection based on scoring functions, either score candidates per token (Dathathri et al., 2019; Lu et al., 2021; 2022) or per reasoning step (Welleck et al., 2022; Uesato et al., 2022; Xie et al., 2024). (ii) Past KV in sequence with tree-structured queries—usually in speculative decoding (Cai et al., 2024; Miao et al., 2023). Further details on these two patterns are discussed in Appendix A.2. Although tree-based search algorithms like A* (Lu et al., 2022) and Monte-Carlo Tree Search (Liu et al., 2023) have been applied, the efficiency of tree-based decoding remains largely under-explored.

**Memory-efficient Attention Algorithms.** Existing memory-efficient attention algorithms target sequence-based decoding. FlashAttention (Dao et al., 2022) improves self-attention computation in LLM training via tiling and kernel fusion, reducing IOs. Flash-Decoding (Dao et al., 2023) extends this, enhancing parallelism by dividing K and V and introducing global reduction to gather partial attention results, enabling efficient decoding for long sequences. Unfortunately, applying these memory-efficient algorithms to the tree-based decoding overlooks redundancy in IO of tree-structured KV cache, which is the focus of DEFT.

**Tree Attention.** Integrated into LLM inference, tree attention reduces computation, storage, and kernel launching overheads (Miao et al., 2023). Tree-structured token candidates undergo parallel decoding, with SpecInfer (Miao et al., 2023) introducing a topology-aware causal masked tree attention algorithm, dynamically updating a causal mask to capture relationships among tokens. Medusa (Cai et al., 2024) uses a similar mechanism with a static causal mask, while other works (Zhao et al., 2023; Liu et al., 2024) adopt analogous approaches to enhance attention calculation efficiency. However, unlike DEFT, these existing works utilizing tree attention do not take memory access into consideration.

**Storage Optimization of Tree-based Decoding.** LLM frameworks optimized for tree-based decoding (Kwon et al., 2023; Zheng et al., 2023) focus on memory storage efficiency. vLLM (Kwon et al., 2023) enhances GPU memory utilization, allowing sequences from the same parent to share KV cache storage. SGLang (Zheng et al., 2023) supports dynamic KV cache management during multi-round interactions with LLMs, improving memory efficiency.

**Discussion on Concurrent Works.** Some concurrent works (Ye et al., 2024a; Juravsky et al., 2024; Athiwaratkun et al., 2024; Ye et al., 2024b; Zhu et al., 2024) also recognize the importance of IO

during LLM inference. However, these works have at least one of these flaws: i) they (Ye et al., 2024a; Juravsky et al., 2024; Athiwaratkun et al., 2024; Ye et al., 2024b; Zhu et al., 2024) cannot be easily extended to situations where the decoding tree has more than two levels—they target single-context batch sampling scenarios, a special case of general tree-based decoding with a system prompt as prefix and unique suffixes in the first depth; ii) they (Juravsky et al., 2024; Athiwaratkun et al., 2024) do not consider the inefficiency caused by the lengths of different nodes in the decoding tree. See the comparison of DEFT and concurrent works in Appendix A.3.

## 3 DEFT

In this section, we first introduce the background knowledge of LLM inference, upon which we outline the importance of QKV partitions for attention calculation. We then present the overview of DEFT algorithm and Attention Kernel design, with its system support. Finally, we propose efficient QKV partitioning method for DEFT, which not only reduces memory access of prefixes' KV cache and partial results (e.g., Softmax), but also ensures balanced partitions during attention computation.

### 3.1 PRELIMINARY

**LLM inference and its bottleneck.** LLM inference involves two stages: (1) prefill and (2) decoding. During the prefill stage, a prompt is tokenized to initialize LLM. The output of the prefill stage becomes the input for the decoding stage. The decoding stage is auto-regressive, with each output token from the previous step serving as the input token for the next step. Due to the sequential process of auto-regressive decoding, LLM inference is memory-bound (Shazeer, 2019; Kim et al., 2023; Cai et al., 2024), wherein every forward pass requires transferring all model parameters and KV cache from slower but larger High-Bandwidth Memory (HBM) to the faster but much smaller shared memory of the GPU (Jia & Van Sandt, 2021) [2]. Another potential bottleneck is low GPU utilization (Dao et al., 2023), which happens when the parallelism (usually limited by the batch size is much smaller than the number of streaming multiprocessors (SMs) on the GPU (108 for an A100), where the operation will only utilize a small portion of the GPU.

**The execution pattern of attention algorithms on GPUs.** We can separate the execution of attention algorithms into two main phases: (1) QKV PREPARATION PHASE: group Query, Key, and Value (QKV) logically to partitions and map QKV groups to different streaming multiprocessors (SMs) of GPUs; (2) ATTENTION CALCULATION PHASE: load QKV partitions to different SMs' shared memory and apply attention algorithms to each group for final attention results.

**QKV partitions with segmented attention.** In sequence-based decoding, QKV partitioning is crucial when the parallelism (usually limited by the batch size (Dao et al., 2023)) is much smaller than the number of streaming multiprocessors (SMs) on the GPU (108 for an A100), where the operation will only utilize a small portion of the GPU. To enable high GPU utilization, Flash-Decoding (Dao et al., 2023) partitions the queries and KV cache then calculates the attention in parallel. Details are as follows: (1) QKV PREPARATION PHASE: for each query in the batch, split its sequential KV cache into chunks as different QKV partitions. (2) ATTENTION CALCULATION PHASE: it calculates segmented attention $A_0$, $A_1$, and $A_2$ over three segments, respectively, and then gets final attention by online Softmax merging (Dao et al., 2022; 2023) based on segmented attention from different QKV partitions. We elaborate on the procedure below.

- Let's say we have key tensor $\boldsymbol{K} \in \mathbb{R}^{l_{kv} \times d}$, value tensor $\boldsymbol{V} \in \mathbb{R}^{l_{kv} \times d}$, and query tensor $\boldsymbol{Q} \in \mathbb{R}^{l_q \times d}$. Considering the general case $\boldsymbol{K}$ and $\boldsymbol{V}$ are partitioned across the sequence (row) dimension into three parts for parallel calculation, respectively: $\boldsymbol{K} = \boldsymbol{K}_0 \parallel \boldsymbol{K}_1 \parallel \boldsymbol{K}_2$, and $\boldsymbol{V} = \boldsymbol{V}_0 \parallel \boldsymbol{V}_1 \parallel \boldsymbol{V}_2$, with "$\parallel$" denoting concatenation along the row axis.
- We calculate the attention $\boldsymbol{A}_0$, $\boldsymbol{A}_1$, and $\boldsymbol{A}_2$ over KV chunks in different streaming-multiprocessors (SMs) of GPU, where $\boldsymbol{A}_0 = \langle \boldsymbol{Q}, \boldsymbol{K}_0, \boldsymbol{V}_0 \rangle$, $\boldsymbol{A}_1 = \langle \boldsymbol{Q}, \boldsymbol{K}_1, \boldsymbol{V}_1 \rangle$, $\boldsymbol{A}_2 = \langle \boldsymbol{Q}, \boldsymbol{K}_2, \boldsymbol{V}_2 \rangle$, and $\langle \mathbf{q}, \mathbf{k}, \boldsymbol{v} \rangle = \text{Softmax}\left(\mathbf{q}\mathbf{k}^\top / \sqrt{d}\right) \boldsymbol{v}$.
- We calculate LogSumExp (LSE) as a weight of merging $\boldsymbol{A}_0$, $\boldsymbol{A}_1$, and $\boldsymbol{A}_2$. We define $\text{LSE}(\mathbf{q}, \mathbf{k}) = \log\left(\sum\left(\exp\left(\mathbf{q}\mathbf{k}^\top / \sqrt{d}\right)\right)\right)$.
- We have $\langle \boldsymbol{Q}, \boldsymbol{K}, \boldsymbol{V} \rangle = \text{SegAttn}(\boldsymbol{A}_0, \boldsymbol{A}_1, \boldsymbol{A}_2)$, which means segmented attention with Online Softmax (Dao et al., 2022):

$$\text{SegAttn}(\boldsymbol{A}_0, \boldsymbol{A}_1, \boldsymbol{A}_2) = \frac{\boldsymbol{A}_0 e^{\text{LSE}(\boldsymbol{Q}, \boldsymbol{K}_0)} + \boldsymbol{A}_1 e^{\text{LSE}(\boldsymbol{Q}, \boldsymbol{K}_1)} + \boldsymbol{A}_2 e^{\text{LSE}(\boldsymbol{Q}, \boldsymbol{K}_2)}}{e^{\text{LSE}(\boldsymbol{Q}, \boldsymbol{K}_0)} + e^{\text{LSE}(\boldsymbol{Q}, \boldsymbol{K}_1)} + e^{\text{LSE}(\boldsymbol{Q}, \boldsymbol{K}_2)}} \, , \text{ where } e := \exp . \quad (1)$$

---

[2] A100's HBM has 1.5-2TB/s bandwidth and 40-80GB; its shared memory has 19TB/s bandwidth and 20MB.

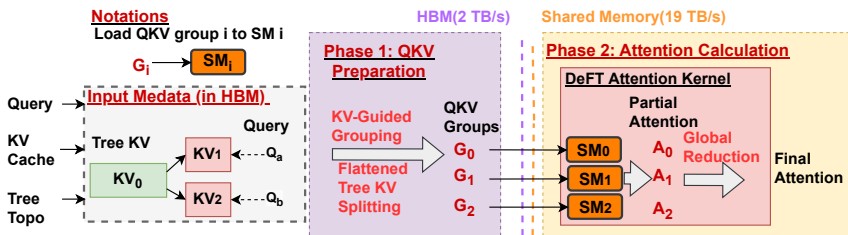

Figure 2: **Overview of DEFT**. *Input Metadata* is prepared in the system elaborated in Appendix A.1. In *QKV Preparation Phase* (see Section 3.3), the QKV will be grouped logically to partitions with IO-awareness of shared prefixes' KV cache and load-balancing. These partitions will guide the loading of QKV on the *Attention Calculation Phase* (see Appendix A.4), where the attention calculation will be executed.

## 3.2 OVERVIEW OF DEFT

**The importance of QKV partitions.** For tree-based decoding, logically partitioning QKV is necessary for attention calculation with high parallelism. The branch number of tree-structured generation requests may be insufficient to fully utilize the GPU when the number of tokens in the tree-structured KV cache is large, due to memory capacity limitations. For example, a request for the reasoning task of sorting 128 numbers (Besta et al., 2023), involves around 40K tokens in a Llama2-7B model, whose KV cache occupies 20GB, which means an 80GB A100 can only process at most 4 requests with such token numbers.

**Motivation of DEFT.** DEFT aims to address two potential bottlenecks (i.e., IO and GPU utilization) of LLM inference when dealing with tree-structured KV sequences. Let's say we have a simple tree with two cascades, as shown in the left part of Figure 2: for two queries $Q_a$ and $Q_b$, the corresponding keys satisfy $K_a = K_0 \parallel K_1$ and $K_b = K_0 \parallel K_2$, respectively, and values obey the same rule. DEFT is designed to: (1) minimize IO by eliminating redundant memory access of the shared prefix's KV cache ($K_0$ and $V_0$) for $Q_a$ and $Q_b$; (2) ensure balanced workloads for high GPU utilization, so that the overhead of computing each segmented attention $A_i$ remains nearly identical. Since the global reduction in equation 1 requires all partial attention, if the overhead for computing $A_i$ is significantly larger than $A_j$, the SM responsible for calculating $A_j$ will experience prolonged idleness.

**Technique overview of DEFT.** DEFT aims to be a hardware-efficient attention algorithm by reducing memory access and ensuring load-balancing for tree-based decoding. See details in Figure 2:
① In the QKV PREPARATION PHASE, for prefix-aware and load-balanced QKV partitions, we introduce a *KV-Guided Grouping* strategy to reuse the KV cache IO of the shared prefixes, and a *Flattened Tree KV Splitting* for high GPU-utilization due to balanced and parallel attention calculation. See details in Section 3.3.
② During the ATTENTION CALCULATION PHASE, we design the DEFT ATTENTION KERNEL[3] to load QKV splits in a memory efficient way, which is logically grouped by the QKV PREPARATION PHASE, then to perform the attention calculation. Key techniques are as follows, with details deferred in Appendix A.4: 1) Common *Kernel Fusion* and *Tiling* strategies avoid significant IO operations for partial results (i.e.. $QK^\top$ and Softmax), which Tree Attention-Medusa (Cai et al., 2024) lacks. 2) *Tree-Topology-Aware Global Reduction*, which extends the global reduction mechanism from Flash-Decoding (Dao et al., 2023). This approach efficiently computes the final attention for each query by aggregating partial attention results from QKV groups while considering the tree structure.

**System frameworks of DEFT.** Apart from efficient DEFT ATTENTION KERNEL, our system for DEFT has other two advantages: 1) efficient memory management of the KV cache in a tree structure, and 2) flexible control of the tree decoding process with arbitrary user-defined functions to decide when and how to branch/prune. The details of key components and their coordinations in the system refer to Appendix A.1.

## 3.3 PREFIX-AWARE AND BALANCED TREE-STRUCTURED KV CACHE PARTITIONS

This section delves into the details of the QKV PREPARATION PHASE, which is a key design aspect of DEFT. The discussion of the ATTENTION CALCULATION PHASE is deferred to Appendix A.4.

---

[3]GPUs utilize a vast array of threads to execute operations known as *kernels*

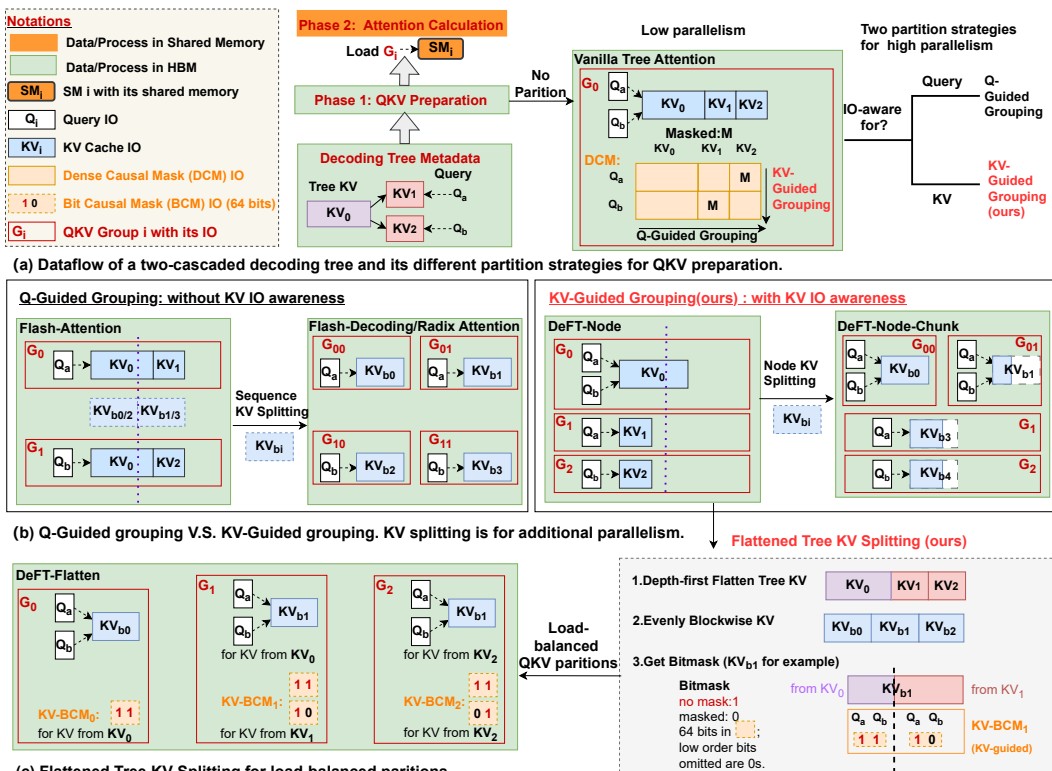

Figure 3: **Comparison of QKV partitioning strategies during the QKV Preparation Phase between DEFT-Node/Node-Chunk/Flatten and different attention algorithm baselines.** Note that the partitioning is logically designed without incurring any data movement costs for QKV. The amount of IO between the GPU HBM and shared memory required by each group is highlighted in red rectangles. Part (a) illustrates the dataflow of a two-cascaded decoding tree example and three categories of QKV partitioning strategies: no partition(Vanilla Tree Attention), Q-Guided Grouping and KV-Guided Grouping. The partitioning strategy will guide the loading of QKV during the subsequent *Attention calculation phase*, where each QKV group $G_i$ will be loaded into $SM_i$ on the GPU. Part (b) shows the comparison of Q-Guided Grouping and KV-Guided Grouping, where the latter can be IO-aware of prefix KV cache $KV_0$ and only load it once. DEFT-Node-Chunk is a weak load-balancing improvement of DEFT-Node by splitting large nodes (e.g., $KV_0$) to chunks. Part (c) illustrates the details (discussed in Remark 3.1) of Flattened Tree KV Splitting in DEFT-Flatten for load-balanced partitions, including Depth-first Flatten strategy, Evenly block-wise strategy, and Bit mask. For a summary of baselines and DEFT, see Table 2. See analysis of tree-attention baselines (Cai et al., 2024; Miao et al., 2023) in Remark 3.2.

Let's begin with a decoding example using the tree-structured KV cache shown in Figure 2. If we group the entire tree-structured KV cache and queries into $G_0$ without any partitions, we can refer to the Vanilla Tree Attention method illustrated in the part (a) of Figure 3. This method calculates attention for all queries simultaneously in a single Streaming Multiprocessor (SM), with the aid of a dense causal mask (DCM).

However, this approach is inefficient due to low GPU utilization, as discussed in Section 3.2. To address this inefficiency, effective partitioning of QKV is essential. This process involves two key considerations: (1) prefix awareness to minimize memory access to the KV cache and (2) load balancing to ensure even distribution of workloads across GPUs.

**Q-Guided vs. KV-Guided Grouping.** Most existing memory-efficient attention algorithms (Dao et al., 2022; 2023; Zheng et al., 2023) adopt *Q-Guided Grouping* for QKV partitioning, where each query serves as the indicator for partitioning, grouping with its corresponding KV cache. However, this method is not prefix-aware, e.g., in Flash-Attention (as shown in Figure 3) $\mathbf{KV_0}$ is loaded twice, namely once for $\mathbf{Q_a}$ and again for $\mathbf{Q_b}$.

We resort to alternative *KV-Guided Grouping* approach: by grouping each node's KV cache with all the queries that share it, the partitioning can be made prefix-aware, therefore reducing memory access to the KV cache. For example, DEFT-Node (shown in Figure 3) only loads the prefix KV

Table 2: **Comparison of QKV partitioning strategies for baselines (most of which are shown in Figure 3) and DEFT.** For IO redundancy, significant issues are highlighted in red, while negligible ones are in blue. "Q" refers to queries, and "KV" refers to the KV cache. "DCM" stands for Dense Causal Mask (a matrix), and "BCM" refers to Bit Causal Mask (a set of 64-bit integers). "PA" represents partial results during attention calculations, including $QK^T$, Softmax, etc. More ⋆ symbols indicate better-balanced workloads for QKV partitions. Details on IO complexity can be found in Appendix A.5.

| Attention Algorithm | Grouping Indicator | KV Split Granularity | IO Redundancy | Load-balancing Level |
|---|---|---|---|---|
| **Flash-Attention (Dao et al., 2022)** | Q-guided | - | KV | ⋆ |
| **Flash-Decoding (Dao et al., 2023)** | Q-guided | by block | KV | ⋆⋆⋆ |
| **Radix Attention (Zheng et al., 2023)** | Q-guided | by block | KV | ⋆⋆⋆ |
| **Tree Attention-S (Miao et al., 2023)** | Q-guided | by block | KV and BCM | ⋆⋆⋆ |
| **Tree Attention-M (Cai et al., 2024)** | entire tree | by GEMM in PyTorch | DCM and PA | ⋆⋆⋆ |
| **Vanilla Tree Attention** | entire tree | no split | DCM and PA | ⋆ |
| **DEFT-Node** | KV-guided | by tree node | Q | ⋆ |
| **DEFT-Node-Chunk** | KV-guided | by tree node, then by block | Q | ⋆⋆ |
| **DEFT-Flatten** | KV-guided | by block | Q and BCM | ⋆⋆⋆ |

cache $KV_0$ once for attention computation. The additional IO cost for queries is negligible since each query only contains a single token, while the KV cache may contain thousands of tokens.

**Tree KV Splitting and Load-Balancing.** Thanks to *KV-Guided Grouping*, DEFT-Node is prefix-aware for KV cache IO. However, it introduces a potential bottleneck: unbalanced workloads across different SMs. For example, as seen in DEFT-Node of Figure 3, $KV_0$ might contain 1,000 tokens, while $KV_1$ only contains 2 tokens. If $G_0$ and $G_1$ are assigned to $SM_0$ and $SM_1$ respectively, $SM_1$ completes computation much earlier and remains idle, leading to low SM utilization[4].

To address this, we need to balance the QKV partitions more evenly. A straightforward approach is to chunk $K_0$, $K_1$, and $K_2$ at the physical level, while maintaining node-wise partitioning at the logical level, as shown in DEFT-Node-Chunk from Figure 3. However, this load-balancing strategy is weak: it only breaks large nodes (e.g., prompts with around 1k tokens) into smaller KV chunks, and it does not handle cases with many small nodes (e.g., speculative decoding), which could slow down inference due to more rounds of GPU execution for additional QKV groups.

As KV cache loading is the primary bottleneck for attention computation (Cai et al., 2024; Tang et al., 2024), it is important to achieve an even token length in each KV cache partition. Therefore, we propose DEFT-Flatten, elaborated in Remark 3.1.

**Remark 3.1** (Techniques of *Flattened Tree KV Splitting*)**.** *As shown in the part (c) of Figure 3, there are three key components:*

- Depth-first Flatten strategy. *This approach minimizes redundant query IO and computation by leveraging the hierarchical relationship between parent and child KV nodes– for instance, queries for parent $KV_0$ (e.g., $Q_a$ and $Q_b$) include those for child $KV_1$ (e.g., $Q_a$). Depth-first flattening instead of breadth-first, maximizes query overlap across KV cache from different nodes but allocated to the same chunk, reducing redundant computations like masked portions in $QK^T$.*
- Evenly block-wise strategy. *It is the core of the splitting, where it ensures equal lengths of KV in each QKV group for balanced workloads of streaming multiprocessors (SMs) in GPUs.*
- Bit mask (Miao et al., 2023). *It is a set of 64-bit integers used to record causal information of tokens in the tree. Therefore, its IO overhead (e.g., two 64-bit integers in KV-BCM$_1$ on part (c) of Figure 3) is negligible compared to the dense causal mask (Cai et al., 2024).*

**Remark 3.2** (Discussion on Tree Attention Algorithms)**.** *Existing attention algorithms are designed for speculative decoding, where attention is calculated for the entire tree-structured queries. However, these methods are not memory-efficient. For partition details, see Figure 11 in Appendix A.5.*

- Tree Attention-Medusa (Cai et al., 2024). *Based on Vanilla Tree Attention (shown on the left in Figure 3), this method uses PyTorch's General Matrix Multiply (GEMM) to partition Q and KV tensors. It is memory-inefficient for two reasons: (1) it does not utilize Flash-Attention to reduce memory access during the computation of intermediate results (e.g., Softmax); (2) it introduces a dense causal mask, whose memory access is significant.*
- Tree Attention-SpecInfer (Miao et al., 2023). *This algorithm employs* Q-Guided Grouping *based on Vanilla Tree Attention and partitions the KV sequence through Flash-Decoding. It is memory-inefficient in redundantly loading the entire tree-structured KV cache for each query.*

---

[4]Considering a microbenchmark that DEFT-Node with 64 queries shares a prompt of 4k tokens, the SM utilization is below 5% for 82.35% time of attention computation, as shown in Table 14.

Table 3: **Comparison of baselines and DEFT.** Attention kernels of baselines are implemented to fit its memory management. Therefore, for a fair comparison with baselines, we implement DEFT-Node and DEFT-Flatten that fit both paged (Kwon et al., 2023)/unpaged memory management.

| Method | Flash-Decoding | Tree Attention-Medusa | Radix Attention(Zheng et al., 2023) | DEFT |
|---|---|---|---|---|
| **Memory Implementation** | unpaged Triton | unpaged PyTorch | paged Triton | unpaged/paged Triton |

Table 4: **Workloads generation**. ToT-BFS stands for Tree-of-Thoughts (Yao et al., 2023) using breadth-first search. APPS (Hendrycks et al., 2021) is a competitive programming problem dataset. Medusa (Cai et al., 2024) is a speculative decoding framework. "GoT" stands for Graph-of-Thoughts (Besta et al., 2023), which contains iteration records using GPT-3.5 for complex reasoning tasks within ToT-BFS. See more details in Table 13.

| Task | Prompt Dataset | Decoding Tree Source | Decoding Tree Collection Method | Stopping Criteria |
|---|---|---|---|---|
| Few-shot prompting | APPS | - | Pad the prompt to 4000 tokens | 400 iterations |
| Multi-step reasoning | 4 tasks in GoT | ToT-BFS | Reconstruct from interaction records with GPT 3.5 in GoT | End of task($\sim$ 3500 iterations) |
| Speculative decoding | APPS | Medusa | Record token tree shape and accepted token length per step | $\sim$1000 steps(max length=6000) |

**IO complexity analysis.** We show DEFT-Flatten is better than existing attention algorithms in IO complexity, including Flash-Decoding (Dao et al., 2023), Tree Attention-Medusa (Cai et al., 2024), and Tree Attention-SpecInfer (Miao et al., 2023). See Appendix A.5.

**Implementation details.** We implement the DEFT attention kernel by OpenAI Triton (Tillet et al., 2019), which enables us to control memory access from global memory to shared memory and attention calculations in a thread block granularity. DEFT-Node and DEFT-Flatten algorithms with two phases in a Python style can be found in Appendix A.8 and Appendix A.9, respectively.

# 4 EXPERIMENTS

In this section, to demonstrate the effectiveness of DEFT under different tree topologies, we comprehensively conduct experiments on three types of tree-based decoding tasks, including: (1) few-shot prompting (Mann et al., 2020): a typical case study of tree-structured interactions with two levels–a prefix and several suffixes; (2) multi-step reasoning (Yao et al., 2023; Xie et al., 2024; Hao et al., 2023): tasks characterized by tree-structured past KV with parallel queries; (3) speculative decoding (Cai et al., 2024; Miao et al., 2023): tasks involving past KV in sequence with tree-structured queries.

## 4.1 EXPERIMENTAL SETUP

**Baselines.** We evaluate the performance of DEFT in NVIDIA A100 (80GB) in Llama3-8B model (Touvron et al., 2023b) with the SOTA attention algorithms in sequence-based and tree-based decoding, as shown in Table 3. Note that we did not include the tree attention operator of SpecInfer (Miao et al., 2023) to our baselines as its kernel only supports at most 64 tokens in the token tree (the decoding tree except for the past sequence KV part), which is unsuitable for tree-based decoding with tree-structured KV (c.f. details in Appendix A.2).

**Workloads generation.** To ensure fairness for workloads of different baselines, we reconstruct decoding trees from real multi-step reasoning and speculative decoding tasks, as shown in Table 4. For multi-step reasoning, we include these four tasks from Besta et al. (2023): (1) Sorting 128 numbers (*Sorting* in short), (2) Document merging (*Document* in short), (3) Keyword counting (*Keyword* in short), and (4) Set intersection (*Set* in short). The tree decoding process would be forced to branch and prune the tree in certain iterations to get the same shape of the decoding tree as the original decoding tree sources. See workload generation details and analysis in Appendix A.6.

## 4.2 ANALYSIS OF MEMORY MANAGEMENT AND BOTTLENECK

As shown in Table 3, the kernel implementations of different attention algorithms adapt to different memory management. To fairly compare their performance of wall-clock time speedup, we need to analyze the influence of memory management and the bottleneck of the system.

**A trade-off between memory storage and memory operation.** In tree-based decoding, storing the KV cache for each branch is simple but lacks shared storage for the prefix's KV cache. Given the limited GPU memory, not accounting for the tree structure in KV sharing reduces the number of tokens the tree can handle. Although storing KV caches by each tree node significantly improves storage efficiency, most attention kernels are designed for sequence-based decoding (Dao et al., 2022;

Table 5: **Comparison of DEFT-Flatten and baselines in average decoding latency (in seconds) for tree-based decoding.** Here, $b$ represents the tree width, and $t$ denotes the token tree size (i.e., the number of tree-structured queries). The fastest method is in **bold**, and the second fastest is underlined. *Radix Attention* is the best baseline in decoding latency. ⋆ denotes out-of-memory (OOM) errors for the A100 80GB GPU. *Speedup Upper-bound (no attention)* refers to the maximum speedup we could achieve for *Radix Attention* if we exclude the attention computation and only run other components including MLP. For more details on attention speedup, see Table 16.

| Memory | Method | Few-shot Prompting | | | Multi-Step Reasoning | | | | Speculative Decoding | | | |
|---|---|---|---|---|---|---|---|---|---|---|---|---|
| | | b=20 | b=30 | b=50 | *Sorting* | *Document* | *Keyword* | *Set* | t=32 | t=64 | t=128 | t=256 |
| Unpaged | Flash-Decoding | 78.96 | 131.19 | 191.09 | 429.65 | 241.20 | 32.75 | 51.76 | 574.50 | 1128.45 | ⋆ | ⋆ |
| | Tree Attention-Medusa | 52.58 | 103.90 | 144.07 | 380.87 | 236.86 | 33.52 | 50.10 | 263.40 | 483.35 | 924.97 | 1881.51 |
| Paged | *Radix Attention* | 12.37 | 14.08 | 16.54 | 104.79 | 69.61 | 11.25 | 17.03 | 54.66 | 69.75 | 108.56 | 188.66 |
| | DEFT-Flatten | **9.98** | **10.99** | **12.48** | **94.67** | **66.95** | **10.90** | **16.10** | **42.23** | **46.60** | **56.96** | **84.27** |
| | *Attention Speedup over Radix Attention* | 1.73× | 1.63× | 1.70× | 1.39× | 1.15× | 1.21× | 1.34× | 1.96× | 2.41× | 3.11× | 3.59× |
| | *Decoding Speedup over Radix Attention* | 1.24× | 1.28× | 1.33× | 1.10× | 1.03× | 1.03× | 1.05× | 1.29× | 1.50× | 1.91× | 2.23× |
| | *Speedup Upper-bound(no attention)* | 1.71× | 2.08× | 2.51× | 1.96× | 1.82× | 1.70× | 1.76× | 1.89× | 2.89× | 3.34× | 4.36× |

Hong et al., 2023; Dao et al., 2023). To use these kernels, KV caches from different nodes must be concatenated into a single tensor, leading to substantial data movement costs (Kwon et al., 2023).

**The benefits of paged memory for tree-based decoding.** For efficient KV cache memory management, paged memory (Kwon et al., 2023; Zheng et al., 2023) is the current mainstream technology. These KV cache tensors are stored in a non-contiguous, paged layout to provide token-level reuse. Besides higher storage efficiency, we note an additional benefit of paged memory management for tree-based decoding: non-contiguous storage in a memory pool is addressed by pointers, ensuring no need to materialize the tree-structured KV into a single tensor before executing the attention kernel. Instead, we only need to record the memory pool addresses of each token's KV cache.

**Bottlenecks and trade-offs.** We provide support for DEFT and baselines with KV cache in memory management (unpaged or paged) according to their designs. We visualize the latency breakdown for (1) KV cache management, (2) attention, and (3) other operations (including MLP calculation) in Figure 4. We observe that with unpaged KV cache management in tree-based decoding, the bottleneck (69.1-83.4%) is the data movement required to materialize the KV cache. However, when we use paged memory management, attention becomes the new bottleneck (51.1-58.3%), especially when the token tree is large.

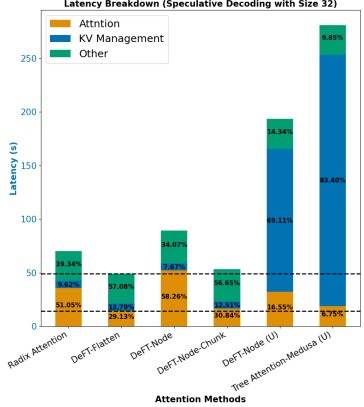

Figure 4: **Latency breakdown for speculative decoding** with a token tree of 32 queries, whose tree topology is from Medusa (Cai et al., 2024). *U* means unpaged memory.

## 4.3 DECODING BEHAVIORS: LATENCY AND IOS.

We evaluate DEFT's performance on various tree-based decoding tasks by measuring decoding latency (Table 5), which demonstrates DEFT's acceleration of tree attention and wall clock time. See attention latency (Table 16), IO (Table 17), and inference accuracy (Table 15) in Appendix A.7. Notably, **decoding latency** essentially represents the optimal end-to-end (e2e) latency, excluding other overheads such as prefill latency (which accounts for approximately 5-10% of e2e latency) and framework-induced overheads (roughly 10-15% of e2e latency), including tree search, branching, etc. We exclude these overheads because they are consistent across all baselines and to eliminate the influence of the framework.

**For few-shot prompting tasks**, we used a prompt with 4k tokens and performed 400 decoding iterations, achieving a 1.33× decoding speedup thanks to 1.70× faster attention calculation and an approximately 90% reduction in IO.

**For speculative decoding tasks**, DEFT-Flatten achieved up to a 2.23× decoding speedup due to up to a 3.59× attention speedup, as the entire token tree (all queries) can share IO of the long prefix.

**For multi-step reasoning tasks**. While DEFT-Flatten improves attention speed by up to 1.36×, the decoding acceleration is less pronounced due to two reasons:(1) The narrow tree width (10) restricts KV cache reuse, though increasing it to 50 in few-shot prompting significantly boosts decoding speed (1.2×-1.5× over 100 iterations, see Appendix A.7). (2)The small number of tokens in the tree keeps attention at 30% of decoding latency, compared to 50-80% in speculative decoding. A longer prompt length increases attention computation overhead, leading to greater speedup, as shown in Table 7.

Table 6: **[Different KV Splitting Strategies]** Comparison of DEFT-Node, DEFT-Node-Chunk and DEFT-Flatten in average attention latency (second) with NVIDIA A100 (80GB) for Llama3-8B model(GQA). This table is supplementary to Table 16. The fastest method is in **bold**, and the second fastest is underlined. Radix Attention is the best baseline in decoding latency. See details of more baselines in Table 16.

| Memory | Method | Few-shot Prompting | | | Multi-Step Reasoning | | | | Speculative Decoding | | | |
|---|---|---|---|---|---|---|---|---|---|---|---|---|
| | | b=20 | b=30 | b=50 | *Sorting* | *Document* | *Keyword* | *Set* | t=32 | t=64 | t=128 | t=256 |
| Paged | Radix Attention | 5.99 | 7.30 | 9.96 | 39.37 | 24.69 | 3.11 | 5.13 | 25.73 | 40.47 | 76.10 | 145.43 |
| | DEFT-Node | 10.59 | 10.62 | 10.85 | 42.96 | 33.29 | 6.16 | 9.58 | 34.59 | 34.41 | 34.96 | 41.78 |
| | DEFT-Node-Chunk | 8.52 | 9.69 | 13.45 | 49.63 | 36.37 | 4.77 | 7.40 | 14.54 | 20.28 | 32.57 | 57.26 |
| | DEFT-Flatten | **3.47** | **4.07** | **5.87** | **28.41** | **21.45** | **2.57** | **3.83** | **13.15** | **16.79** | **24.46** | **40.56** |

Table 7: **[Different Prompt Lengths]** Comparison of DEFT-Flatten and Radix Attention in the efficiency of multi-step reasoning task `sorting`. The original prompt length is approximately 1K tokens, and we pad it to lengths of 5K, 8K, or 10K tokens.

| Speedup | Prompt length L | | | |
|---|---|---|---|---|
| | L=1k | L=5k | L=8k | L=10k |
| **Attention** | 1.39× | 1.71× | 1.97× | 1.84× |
| **Decoding** | 1.09× | 1.37× | 1.53× | 1.67× |

Table 8: **[Different Model Sizes]** Comparison of decoding latency speedup and Attention/FFN latency ratio (in short as **A/F-LR**) between DEFT and Radix Attention for Codellama-34B and Codellama-7B models. Radix Attention is the best baseline in decoding latency. $b$ represents the tree width, and $t$ denotes the token tree size. For multi-step reasoning, we test the task *sorting* whose prompt length is about 1k tokens.

| Metric | Model Size | Few-shot Prompting | Multi-step Reasoning | Speculative Decoding |
|---|---|---|---|---|
| | | b=30 | *Sorting* | t=64 |
| Decoding Time | 7B | 1.34× | 1.09× | 1.85× |
| Speedup | 34B | 1.23× | 1.03× | 1.78× |
| Radix Attention's | 7B | 1.27 | 1.12 | 2.12 |
| A/F-LR | 34B | 0.80 | 0.48 | 1.66 |
| DEFT-Flatten's | 7B | 0.68 | 0.89 | 0.69 |
| A/F-LR | 34B | 0.45 | 0.42 | 0.49 |

## 4.4 ABLATION STUDY

We evaluate the influence of different KV splitting strategies, model sizes, and prompt lengths in this subsection. See more ablations in Appendix A.6, including the influence of different GPUs (Table 19), chunk sizes during KV splitting (Figure 15), and model architectures (Table 20 and Table 21).

**The impact of KV splitting strategy in DEFT.** We compared three KV splitting strategies with the baseline Radix Attention, as shown in Table 6. DEFT-Flatten consistently outperforms the others across all tree-structured settings. DEFT-Node-Chunk generally performs better than DEFT-Node because it splits large nodes into smaller chunks for more balanced computations, especially when $b \leq 30$ and $t \leq 64$, as well as in reasoning tasks like Keyword and Set. However, it struggles with many small nodes (e.g., prompts with around 1k tokens), leading to slower inference due to more GPU execution rounds required for additional QKV groups (see $t = 256$ for DEFT-Node-Chunk).

**The influence of prompt length.** See Table 7. With a longer prompt, DEFT-Flatten shows a more pronounced speedup in the same model, since the attention overhead is proportional to the token count in the decoding tree, while the FFN overhead remains nearly constant for the same model. See Figure 16, Figure 17 and Figure 18 for more results.

**The influence of model size.** See Table 8. With the larger model, Codellama-34B, DEFT-Flatten achieves slightly reduced but significant (up to 1.78×) decoding speedup. The performance reduction is attributed to a lower A/F-LR, as the FFN overhead is greater due to larger hidden dimensions.

## 5 CONCLUSION

We propose DEFT-Flatten, a hardware-efficient attention algorithm optimized for tree-structured LLM inference. It effectively addresses memory access and GPU utilization bottlenecks by reusing shared prefixes' KV cache and evenly distributing workload across partitions. DEFT-Flatten's key strengths lie in its prefix-sharing awareness and load balancing, making it versatile for various tree-structured tasks. It also scales well with larger search spaces and multiple branches. Our results show that DEFT-Flatten achieves up to 2.23×/3.59× speedup in decoding and attention latency, outperforming baselines in tasks such as few-shot prompting, multi-step reasoning, and speculative decoding. Our ablation studies highlight that: (1) balanced partitioning is critical, (2) DEFT-Flatten delivers significant speedups across various LLM models and GPU architectures, and (3) DEFT-Flatten provides even greater speedups with larger token sizes (e.g., longer prompts) and more branches in tree-structured requests.

ACKNOWLEDGEMENT

This work was supported in part by the National Science and Technology Major Project (No. 2022ZD0115101), Research Center for Industries of the Future (RCIF) at Westlake University, Westlake Education Foundation, and Westlake University Center for High-performance Computing.

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

CONTENTS OF APPENDIX

# A APPENDIX

## A.1 COMPONENTS OF SYSTEM SUPPORT FOR DEFT

The left part of Figure 5 shows the coordination of different components for efficient and flexible tree-based decoding. The details of functions for system components of DEFT are as below:

1. **Branch Controller**: It makes the tree decoding process forced by a user-defined function (e.g. branch to two children every 3 iterations, as the example shown in the right of Figure 5). Tree-search-based algorithms can be applied here using the decoding tree's topology information.

2. **Sequence Tree Manager**: It maintains the topology of the decoding tree based on the tree operations and tokens from the Branch Controller. The tree operations like pruning and branching will be executed by *Tree Handler* in this component. *Branch Result Storage* will record token generation results of all branches in the decoding tree, and output when the decoding stops.

3. **KV cache Manager**: It will maintain KV cache with a tree structure. A map between sequence IDs in the decoding tree and KV cache index is kept, which will be updated based on KV operations[5] from the Sequence Tree Manager. We provide both paged (Kwon et al., 2023) and unpaged memory management in this part to fit different attention kernels.

4. **Model Interface**: pass input metadata to DEFT Attention kernel and MLP module, then return logits and memory pointers of updated KV cache.

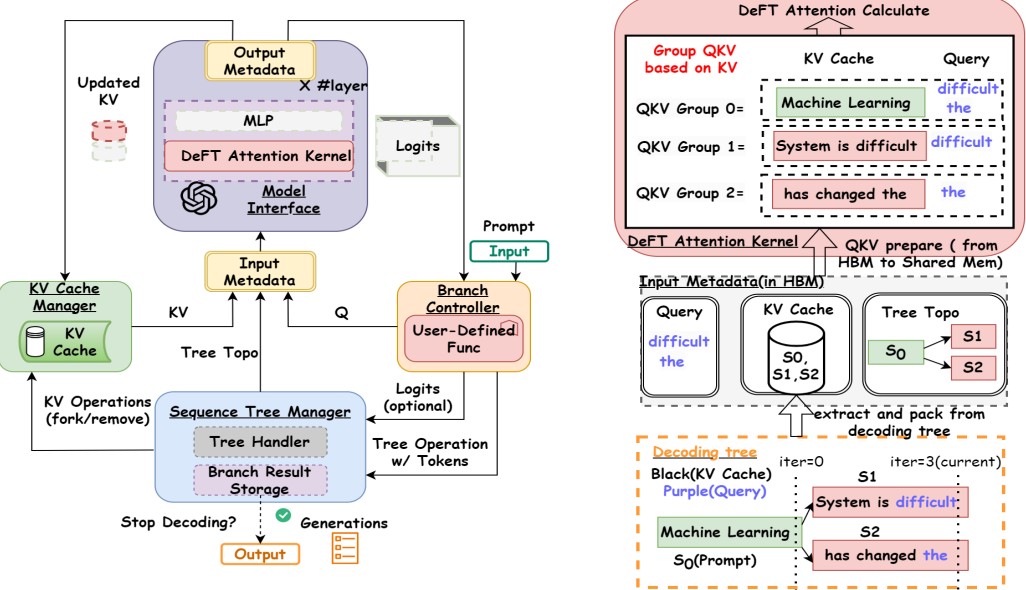

Figure 5: **Illustration of DEFT**. (Left) System overview. (Right) The data flow of DEFT-Node (DEFT-Flatten is similar except for QKV partitioning) using a decoding tree example.

The right part of Figure 5 further showcases the key data flow of the system through a decoding tree example. For simplicity, we present DEFT-Node here and DEFT-Flatten is similar except for QKV

---

[5]e.g. when a node is pruned in the decoding tree, its KV space will be evicted using a *Remove* operation.

partitioning. Input metadata will be extracted by three components we mentioned above, then loaded from HBM to shared memory in a group manner after the QKV PREPARATION PHASE discussed in Section 3.3. Then QKV groups will be processed by DEFT ATTENTION KERNEL in ATTENTION CALCULATION PHASE of DEFT. See details of techniques in these two phases in Appendix A.4.

## A.2 DISCUSSION OF TREE-BASED DECODING

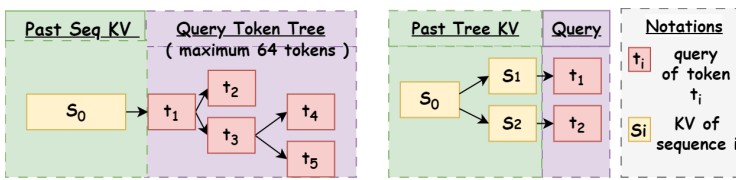

(a) (Left) Sequence KV with queries in a tree for parallel decoding (Miao et al., 2023; Cai et al., 2024), where a *causal mask* is applied to record the causal information among queries in a tree of tokens. (Right) Tree KV with parallel queries for shared prefixes in multi-step reasoning.

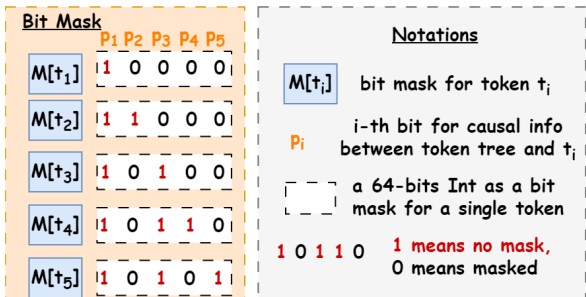

(b) *Bit Mask* in SpecInfer (Miao et al., 2023) to record the causal information between query tokens in a tree structure. The decoding tree is in the left part of 6a.

Figure 6: Discussion of **tree-based decoding** with tree queries (Miao et al., 2023) and tree KV.

Tree-based decoding could have tree-structured KV cache for storage with awareness of shared prefixes (Zheng et al., 2023), or tree-structured queries in parallel/speculative decoding (Miao et al., 2023; Cai et al., 2024), as shown in Figure 6. A general decoding could both do with tree KV and tree queries, which could reduce redundancy (e.g. IO, storage, computation, etc) of shared prefixes, as well as increase the generated tokens per decoding iteration.

The existing inference frameworks (Zheng et al., 2023; Gim et al., 2023) focused on tree-based decoding efficiency primarily aim to: (1) reduce memory footprints (Zheng et al., 2023) to enable larger batch sizes for higher throughput; (2) reuse the prompt cache (Gim et al., 2023) to avoid recomputation of the KV cache for faster time-to-first-token (TTFT). However, their designs do not specifically target reducing the latency of the entire decoding process. We observe that the tree-structured feature of LLM inference could provide us with some advantages to speed up the decoding itself.

**Analysis of speedup potential in tree-based decoding.** In tree-based decoding, KV cache and queries can be structured in a tree. Not only can we store KV cache in a tree, but also we can load QKV with awareness of tree topology during attention calculation, to minimize the expensive IO between HBM and on-chip shared memory of GPUs. We explain it in two case studies of complex scenarios with tree-structured interactions: (1) multi-step reasoning (Yao et al., 2023; Xie et al., 2024); (2) speculative decoding (Cai et al., 2024; Miao et al., 2023).

**Case study 1: multi-step reasoning.** As shown in the left part of Figure 7, we can summarize process of multi-step reasoning (Hao et al., 2023; Yao et al., 2023; Besta et al., 2023) to three phases: (1) *Thought Generation*: generate k candidates for the next thought step based on a generation prompt $P_g$ and previous steps $S$; (2) *Thought Evaluation*: When presented with a frontier of various thoughts, a LLM as state evaluator measures previous thoughts $S$ based on an evaluation prompt $P_e$ towards resolving the problem. This assessment acts as a heuristic for the search algorithm, guiding it on

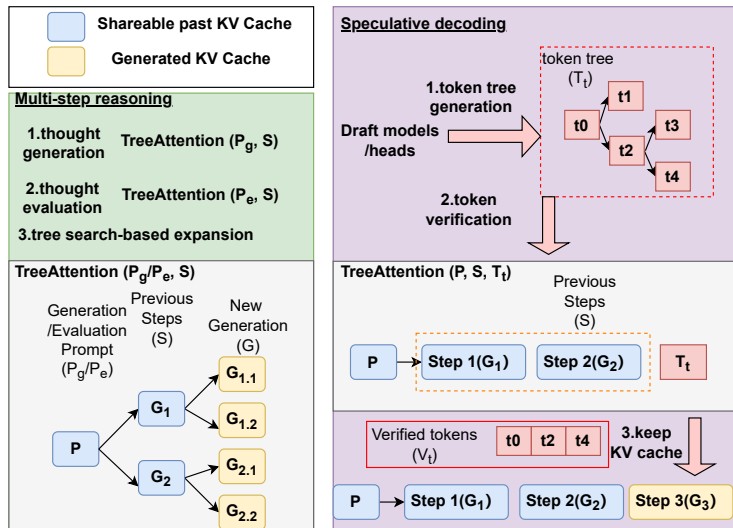

Figure 7: Analysis for two case studies of tree-based decoding. (Left) Multi-step reasoning. (Right) Speculative decoding. Blue boxes mean shareable past KV cache in storage and memory access during the tree attention calculation, while yellow boxes mean the KV cache of generated context.

which states to pursue further and the sequence in which to explore them; (3) *Tree Search-based Expansion*: play different search algorithms (Lu et al., 2022; Liu et al., 2023; Xie et al., 2024) to explore search space, which influences the future tree topology. In both (1) and (2), we can share IO of KV cache for $P_g/P_e$ and $S$ during tree attention calculation.

**Case study 2: speculative decoding.** As shown in the right part of Figure 7, we can summarize the process of speculative decoding (Cai et al., 2024; Miao et al., 2023) to tree phases: (1) *Token Tree Generation*: multiple small draft models (Miao et al., 2023) or fine-tuned heads (Cai et al., 2024) generate multiple sequences of tokens based on prompt $P$, then they are merged to a speculated token tree $T_t$, which is very fast (e.g. $1\%$ of time overhead in SpecInfer (Miao et al., 2023)); (2) *Token Verification*: based on these tree-structured token candidates $T_t$, verify the correctness of its tokens against an LLM's output, where tree-attention calculation is the bottleneck of the process (Miao et al., 2023). In (2), we can share IO of KV cache for $P$ and $S$ during tree attention calculation.

**Why existing tree-attention algorithms are not enough?** The existing tree-attention algorithms are either in-efficient in memory access (Cai et al., 2024; Miao et al., 2023) or not suitable for general tree-based decoding (Miao et al., 2023) with more than 64 tokens in the token tree.

- In SpecInfer(Miao et al., 2023), as shown in Figure 6b, a *bit mask* is utilized to record the causal information among queries of a token tree. Each token $t_i$ in queries will have a 64-bit Int as a *bit mask*, where j-th bit means the causal relationship between query of $t_i$ and KV cache of $t_j$. The advantage of this mask design is that it greatly reduces IO, but it results in the maximum number of tree tokens being only 64, which is not practical for scenarios with tree-structured KV cache. What's more, it is not IO-aware for KV cache as it will load KV cache of the entire tree for each query.
- Medusa (Cai et al., 2024) is suitable for general tree-based decoding, but it is not hardware-efficient due to significant IOs of a dense causal mask and partial results during attention calculation (e.g. softmax).

## A.3 DISCUSSION OF CONCURRENT WORKS

There are some concurrent works (Athiwaratkun et al., 2024; Ye et al., 2024a; Juravsky et al., 2024; Zhu et al., 2024) in attention algorithm design for single-context large-batch sampling, where the goal is to generate multiple sequences from a single context(e.g. system prompt or few-shot examples), which is a special case of tree-based decoding with a depth of 1. The design of their algorithms are

based on this feature, which means they can not suit well in attention calculation of a tree with more than two levels of prefixes with efficiency.

**Insights and techniques in common.** Both concurrent works and DEFT have the insight that memory access is the bottleneck of LLM inference, and decomposing attention across subsequences to reduce the memory access of the prefix KV: (1) calculate attention $A_p$, $A_s$ over prefix and suffixes, respectively; (2) get finial attention by online softmax merging (Dao et al., 2022; 2023) based on $A_p$ and $A_s$. Here are the details of the correctness proof:

- Let's say we have key tensor $K \in R^{(l_{kv},d)}$, value tensor $V \in R^{(l_{kv},d)}$, and query tensor $Q \in R^{(l_q,d)}$. Consider the general case K and V are partitioned across the sequence (row) dimension into two parts for prefix and suffixes, respectively: $K = K_p \parallel K_s$, and $V = V_p \parallel V_s$, with $\parallel$ denoting concatenation along the row axis.
- We calculate the attention $A_p$, $A_s$ over prefix and suffixes, where $A_p = \langle Q, K_p, V_p \rangle$, $A_s = \langle Q, K_s, V_s \rangle$, and $\langle q, k, v \rangle = \text{Softmax}\left(\frac{qk^T}{\sqrt{d}}\right) v$.
- Based on Equation 1, we can have segmented attention $\langle \boldsymbol{Q}, \boldsymbol{K}, \boldsymbol{V} \rangle = \text{SegAttn}(\boldsymbol{A}_p, \boldsymbol{A}_s)$.

Table 9: Comparison among DEFT and concurrent works in single-context large-batch sampling scenarios, including Chunk-Attention (Ye et al., 2024a), Hygragen (Juravsky et al., 2024) and Bifurcated-Attention (Athiwaratkun et al., 2024). RelayAttention (Zhu et al., 2024) and Cascade-inference (Ye et al., 2024b) are similar to Hygragen. More $\star$ means more balanced workloads after tree split, which also shows how insensitive the acceleration is to the tree topology.

| Method | Chunk-Attention | Hygragen | Bifurcated-Attention | DEFT-Node | DEFT-Node-Chunk | DEFT-Flatten |
|---|---|---|---|---|---|---|
| **IO-aware levels** | 2 (depth $\leq$ 1) | 2 (depth $\leq$ 1) | 2 (depth $\leq$ 1) | all (every depth) | all (every depth) | all (every depth) |
| **Tree KV split granularity** | by node first, then by block | by tree depth | by tree depth | by tree node | by tree node, then by block | flatten tree, then by block |
| **Load-balanced level** | $\star\star\star$ | $\star\star$ | $\star\star$ | $\star$ | $\star\star\star$ | $\star\star\star\star$ |
| **Goal metrics** | throughput | throughput | latency | latency | latency | latency |

**Comparison of differences.** The existing works of single-context large-batch sampling are not hardware-efficient for general tree-based decoding for two reasons, as shown in Table 9:

- They are designed for decoding trees with only two levels—prefixes at the root and suffixes at depth 1. For decoding trees with multiple levels of prefixes, their algorithm can only reduce the IO of the prompt at the root of the tree. However, in scenarios such as multi-step reasoning (Yao et al., 2023; Besta et al., 2023; Hao et al., 2023), the token length of non-root prefixes can also be very long (e.g., thousands of tokens), and their KV cache's IO is not reused. DEFT can reuse KV IO of all non-leaf prefixes in a general decoding tree, providing greater acceleration potential.
- They have not addressed the unbalanced workload problem in tree-based decoding. Nodes in the decoding tree can vary significantly, making it crucial to split the tree and group QKV in a way that ensures balanced calculations for each QKV group. Simply dividing based on depth alone is insufficient. For example, in speculative decoding, the prefix might contain thousands of tokens, while each layer only processes a few dozen tokens (Cai et al., 2024; Miao et al., 2023).

## A.4 DISCUSSION OF TECHNIQUES IN EFFICIENT ATTENTION ALGORITHM DESIGN

Table 10: Technique list of DEFT. What we propose is in red. The details of the first four techniques are in Section 3.3, while the details of the following techniques are discussed in this chapter.

| Technique | Goal |
|---|---|
| *KV-Guided Grouping* | High utilization of GPU and minimal KV cache IO between HBM and shared memory. |
| *Flattened Tree KV Splitting* | Balanced attention calculation for high computation efficiency. |
| *Bit Causal Mask* (Miao et al., 2023) | Record causal information of tokens in the decoding tree with little IO cost. |
| *Kernel Fusion* (Dao et al., 2022; 2023) | Reduce partial results IO (e.g. $\boldsymbol{QK}^T$, Mask $M$, and Softmax, etc ). |
| *Tiling* (Dao et al., 2022; 2023) | Enable attention calculation within limited size of GPU's shared memory. |
| *Tree-topology Aware Global Reduction* | To get the correct tree attention of the entire decoding tree. |

In this subsection, we summarize and discuss the common techniques in existing designs of efficient attention algorithms and kernels : (1) *Kernel Fusion* with *Tiling* strategy (Dao et al., 2022; Hong et al., 2023; Miao et al., 2023); (2) *Tree-topology Aware Causal Mask* (Miao et al., 2023; Cai et al., 2024); (3) *KV Split* with *Global Reduction*(Hong et al., 2023). Then we explain the details of design in DEFT Attention Kernel, where the techniques are in Table 10.

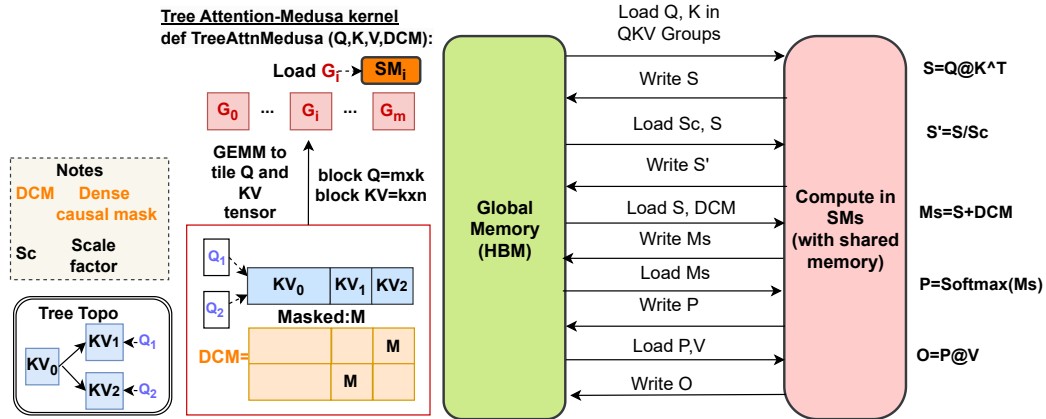

Figure 8: Operations of Tree Attention-Medusa (Cai et al., 2024). No *Kernel Fusion* or *Tiling* strategy is applied, which introduces significant IO of partial results like $QK^\top$, DCM, and Softmax between GPU global memory and on-chip shared memory.

*Kernel Fusion* is a common technique of IO reduction: if multiple operations are performed on the same input, it is more efficient to load the input once from HBM rather than loading it multiple times for each operation; Similarly, the same principle applies when transferring output from shared memory to HBM. To fuse all the attention operations into one GPU kernel with the limited size of shared memory, we further utilize the commonly employed *Tiling* strategy (Dao et al., 2022; 2023): split queries and KV cache within each QKV group to small blocks to prevent materialization of attention matrix in HBM by computing attention within the limited size of shared memory, then incrementally performing the softmax reduction as the formulation in Equation 1 to reconstruct the attention.

**Remark A.1** (Importance of tiling and fused kernel during ATTENTION CALCULATION PHASE). *Methods in this phase can be roughly divided into two categories: (1) without tiling and kernel fusion: Tree Attention in Medusa (Cai et al., 2024), which introduces significant IO operations for partial results (i.e.. $QK^\top$ and Softmax), as shown in Figure 8; (2) with tiling and a fused kernel: Flash Decoding (Dao et al., 2023), Tree Attention in SpecInfer (Miao et al., 2023) and our DEFT.*

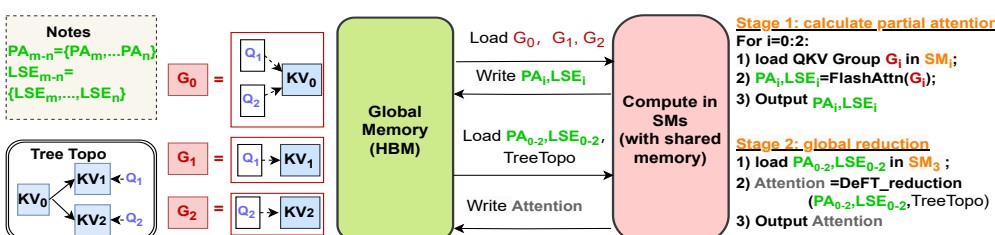

Figure 9: *Overview of two stages in* DEFT *Attention Kernel (*DEFT*-Node for example, and* DEFT*-Flatten is similar).* **Stage 1–calculate partial attentions.** Based on the QKV grouping results after *KV-Guided Grouping Strategy with Tree Split* as mentioned above, each QKV group ($G_i$) will be allocated to a thread block for Flash Attention (Dao et al., 2022) calculation with common *Kernel Fusion* and *Tiling* strategy. Similar to Flash-Decoding (Dao et al., 2023), we not only get partial attention ($PA_i$) but also return "LogSumExp" ($LSE_i$) as a weight parameter for the next stage's reduction. **Stage 2–global reduction.** Upon receiving $PA_i$ and $LSE_i$ for each QKV group $G_i$, DEFT now performs a *Tree-Topology-Aware Global Reduction* ($DeFT\_reduction$). Guided by the tree topology among sequence nodes of KV in the decoding tree, DEFT logically remaps the partial results of attention and LogSumExp to get the correct final attention for each query after reduction. The decoding tree is the same as the one in the left of Figure 3. $SM_i$ means the streaming multiprocessor $i$ in GPU.

The *Tree-topology Aware Causal Mask* (*Causal Mask* for short) is introduced in speculative decoding works (Miao et al., 2023; Cai et al., 2024) to facilitate the calculation of attention for all queries within a decoding tree using a single GPU kernel. It achieves this by recording the causal relationships among queries and KV cache in the decoding tree. As depicted in Figure 6, while originally designed for tree-based decoding with KV cache for a sequence of tokens and tree-structured queries, the *Causal*

*Mask* can also be adapted to tree decoding with tree-structured KV cache and parallel queries—a configuration targeted by DEFT to enhance efficiency.

**Remark A.2** (The effects of introducing a causal mask). *Causal mask brings two parts of redundancy:*

- *Memory Access. Medusa (Cai et al., 2024) materializes the dense causal mask (DCM) in HBM to record the causal information between $n_q$ tokens in queries and $n_{kv}$ tokens in the KV cache, thereby introducing a significant IO cost for loading this $n_q \times n_{kv}$-sized mask to shared memory. SpecInfer (Miao et al., 2023) introduces a 64-bit integer as a bit causal mask (BCM) to record the causal information among up to 64 tokens, which incurs minimal IO cost from HBM to shared memory but is not suitable for decoding trees with more than 64 tokens. Details regarding the design of the bit mask in SpecInfer are discussed in Appendix A.2.*
- *Computation. In addition to the computational cost of generating the causal mask itself, there is an additional redundancy in computation: many of the matrix multiplication results of $\boldsymbol{QK}^\top$ are masked out and never utilized. Both Medusa and SpecInfer have this issue.*

DEFT-*Flatten in Appendix A.9 adopts a bit causal mask insipred by SpecInfer (Miao et al., 2023) to minimize the IO of the causal mask. Details of the bit mask design are on the left of Figure 3.*

*Splitting* is introduced to improve GPU utilization in sequence-based decoding (Hong et al., 2023), which is necessary when the parallelism is limited by a small batch size for long-context scenarios. Flash-Decoding splits long KV and group QKV based on Q first, then these groups will be allocated to different streaming multi-processors (SMs) in the GPU to get partial attention via Flash Attention (Dao et al., 2022).

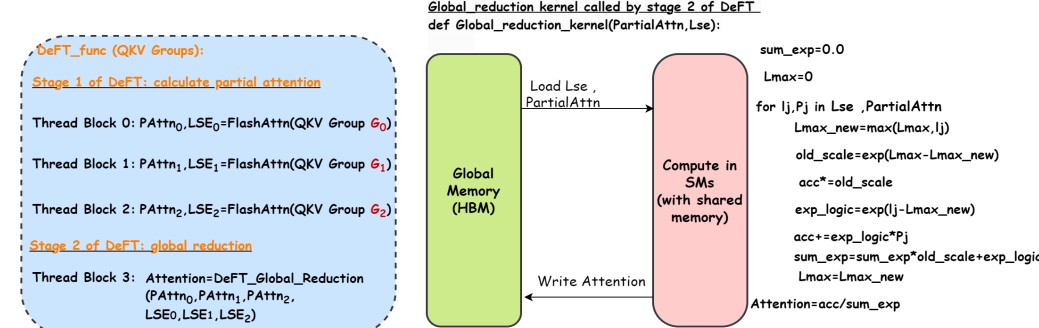

(a) Left: Illustration of DEFT Attention Kernel with two stages. Right: Global reduction kernel called in DEFT stage 2 illustrated in Figure 10b. QKV Groups $G_0$, $G_1$ and $G_2$ are from DEFT QKV groups in Figure 3.

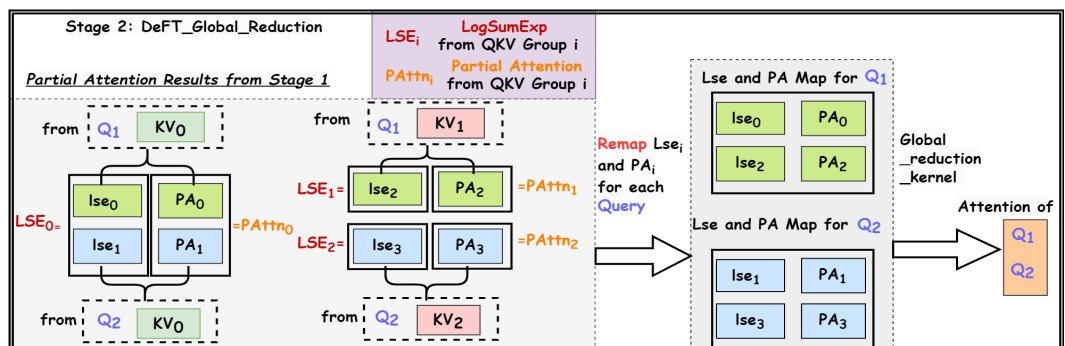

(b) Stage 2 of DEFT: Global Reduction (DEFT-Node for example). Based on tree topology in Figure 3, we can group LogSumExp and Partial Attention based on Query, then we call the Global reduction kernel in the right of Figure 10a to get the final attention.

Figure 10: **Detailed attention operations of DEFT kernel (DEFT-Node for example, and DEFT-Flatten is similar)**. Based on the same decoding tree in Figure 3.

To obtain the accurate final attention, partial attentions from QKV groups with identical queries need to be grouped for *Global Reduction*.

Similarly, DEFT also splits the decoding tree to different QKV groups for balanced workloads of SMs in the GPU, which is the *Flattened Tree KV Splitting* strategy we propose in Section 3.3, as illustrated in the bottom right part of Figure 3. To obtain the correct tree attention, DEFT also requires a global reduction. However, the global reduction in Flash-Decoding is for sequence-based decoding, which cannot aware the tree-topology for global reduction in tree-based decoding. Therefore, we propose *Tree-Topology-Aware Global Reduction*, as shown in the Figure 10b.

Based on the techniques mentioned above, we designed the DEFT Attention Kernel with two stages, as shown in Figure 9, to execute the attention operations after the **QKV Preparation Phase** of DEFT, which we elaborated on in Section 3.3. For more details on the DEFT Attention Kernel, see Figure 10. The attention operations of DEFT-Flatten are omitted because they are very similar to those of DEFT-Node, except for the usage of the bit causal mask for tree attention calculation.

## A.5 ANALYSIS: IO COMPLEXITY OF DEFT

This section analyzes the IO complexity of DEFT, showing a significant reduction in HBM accesses compared to existing attention algorithms. Note that it is non-trivial to summarize the IO cost of the entire tree decoding process, thus we only compare IOs based on the decoding tree snapshot in a single iteration.

Consider a decoding tree with the features outlined in Table 11, and we summarize the corresponding IO breakdown in Table 12.

Table 11: **Notations**.

| | |
|---|---|
| $l_n$ | Number of leaf nodes in a decoding tree, which means how many queries are in this decoding iteration. |
| $N_i$ | Total token length from the root node to leaf node i. |
| $N_{tree}$ | Total token length the entire tree. |
| $\#node$ | Total number of nodes in entire tree. |
| $n_i$ | The token length of node i. |
| $d_{head}$ | Head dimension of LLM. |
| $s_c$ | Scale factor for scaled dot-product attention, typically denoted as $\sqrt{d_{\text{head}}}$. |
| $F_s$ | Shared factor of reusing prefixes in tree attention, which means to which extent we can reduce IOs of KV cache: $F_s = (\sum_{i=1}^{ln} N_i)/N_{tree}$. |

It can be observed that *due to the lack of tree-topology awareness, sequence-based decoding methods, such as naive attention and Flash-Decoding, incur $F_s$ times more memory access overheads for KV cache compared to* DEFT-*Node/Node-Chunk/Flatten and Tree Attention-Medusa (Cai et al., 2024).*

However, Tree Attention-Medusa entails higher IO overheads for partial results like $QK^\top$ and Softmax due to the lack of tiling and kernel fusion[6]. What's more, a dense mask is introduced to record the causal information of tokens in the tree, with significant IO costs, as shown in the left of Figure 11.

When the number of leaf nodes/queries $ln$ is sufficiently large, the IO cost of partial results might become comparable to that of the KV cache. For instance, in the Llama models (Touvron et al., 2023a;b), where $d_{head} = 128$, with $l_n = 29$, the total IO cost of $QK^T$, $M$, $\frac{QK^\top}{s_c}$, $M + \frac{QK^\top}{s_c}$, and Softmax matches that of the KV cache.

**Remark A.3** (KV IO in SpecInfer). *Though similar to* DEFT, *SpecInfer (Miao et al., 2023) also employs a fused kernel for tree attention. As shown in Figure 11, SpecInfer adopts* Q-Guided Grouping. *Therefore, no IO is sharing for KV cache among queries in SpecInfer: instead, each query will load the entire KV cache of the tree independently, bringing significant IOs of the KV cache as in Table 12.*

---

[6]Note that $QK^T$, $\frac{QK^\top}{s_c}$, $M + \frac{QK^\top}{s_c}$ and Softmax will load and write, so the IO cost contains a round-trip of memory access between HBM and shared memory, as shown in Figure 8.

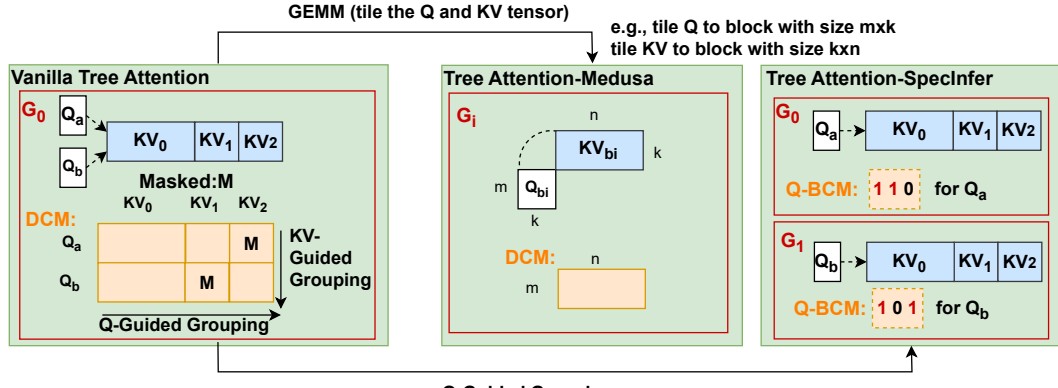

Figure 11: **(Supplementary to Figure 3) QKV partitioning of Tree Attention (Cai et al., 2024; Miao et al., 2023) and memory access.** Tree Attention-Medusa (Cai et al., 2024) partitions QKV by General Matrix Multiply(GEMM) in PyTorch. Tree Attention-SpecInfer (Miao et al., 2023) adopts Q-Guided Grouping. Q-$BCM$ is the Q-Guided bit causal mask for SpecInfer, where each bit represents the causal relationship between a query and a segment of tokens in the KV cache. For example,Q-$BCM$ for $Q_a$ is "110" which means the first two segments of KV cache $KV_0$ and $KV_1$ is valid for $Q_a$'s attention. The $Q_i$ and $K_j$ in the figure are the same as the ones in Figure 3.

Table 12: **IO complexity breakdown for various methods.** $\mathcal{O}(1)$ denotes the IO cost for a single data in the tensor across all layers and heads, which is equivalent to $\#heads * \#layer * dtype\_size$. The best among all methods in the table is in red, while the (potential) worst is in blue. Query IO is omitted as it is $\mathcal{O}(kl_nd_{head})$ for all methods. Here, $k$ is the number of QKV groups: for DEFT-Node $k = \#node$;for DEFT-Node-Chunk $k = \sum_{i=1}^{\#node} ceil(n_i/b_s)$, which is the node number after chunk wise; for DEFT-Flatten, $k = Ntree/b_s$, where $b_s$ is the block size of KV; for others, $k = 1$. M in Tree Attention-M is short for Medusa (Cai et al., 2024), while S in Tree Attention-S is short for SpecInfer (Miao et al., 2023).

| Method | KV cache | $QK^\top$ | $\frac{QK^\top}{s_c}$ | Mask(M) | $M + \frac{QK^\top}{s_c}$ | Softmax |
|---|---|---|---|---|---|---|
| Naive Attention | $\mathcal{O}(2d_{head}\sum_{i=1}^{l_n} N_i)$ | $\mathcal{O}(2\sum_{i=1}^{l_n} N_i)$ | $\mathcal{O}(2\sum_{i=1}^{l_n} N_i)$ | 0 | 0 | $\mathcal{O}(2\sum_{i=1}^{l_n} N_i)$ |
| Flash-Decoding | $\mathcal{O}(2d_{head}\sum_{i=1}^{l_n} N_i)$ | 0 | 0 | 0 | 0 | 0 |
| Radix-Attention | $\mathcal{O}(2d_{head}\sum_{i=1}^{l_n} N_i)$ | 0 | 0 | 0 | 0 | 0 |
| Tree Attention-M | $\mathcal{O}(2d_{head}N_{tree})$ | $\mathcal{O}(2l_nN_{tree})$ | $\mathcal{O}(2l_nN_{tree})$ | $\mathcal{O}(l_nN_{tree})$ | $\mathcal{O}(2l_nN_{tree})$ | $\mathcal{O}(2l_nN_{tree})$ |
| Tree Attention-S | $\mathcal{O}(2d_{head}N_{tree}l_n)$ | 0 | 0 | $\mathcal{O}(l_nN_{tree}/64)$ | 0 | 0 |
| DEFT-Node | $\mathcal{O}(2d_{head}N_{tree})$ | 0 | 0 | 0 | 0 | 0 |
| DEFT-Node-Chunk | $\mathcal{O}(2d_{head}N_{tree})$ | 0 | 0 | 0 | 0 | 0 |
| DEFT-Flatten | $\mathcal{O}(2d_{head}N_{tree})$ | 0 | 0 | $\mathcal{O}(N_{tree})$ | 0 | 0 |

**Remark A.4** (IO in Radix Attention). *Radix Attention (Zheng et al., 2023) is essentially an implementation of Flash-Decoding (Dao et al., 2023) utilizing paged and tree-structured memory management. As a result, the IO behavior is identical to that of Flash-Decoding, as shown in Table 12.*

**Remark A.5** (Causal mask IO). DEFT-*Node splits the decoding tree by nodes without the need for causal masks. For more balanced calculations among SMs in GPUs,* DEFT-*Flatten evenly splits the decoding tree into blocks, with minimal IO cost for masks inspired by SpecInfer. This design reduces the IO overhead of masks significantly compared to the dense mask design in Medusa, as shown in Table 12.*

### A.6 DISCUSSION OF WORKLOADS GENERATION

**The rationality of workload settings.** To validate DEFT's acceleration across various decoding tree topologies, we compiled decoding trees from real tasks, covering the following three aspects:

- Few-shot prompting: This involves a two-level tree with a prompt prefix and multiple branches for suffix generation. As a case study, we fixed the prompt length at approximately 4000 tokens and varied the number of branches.

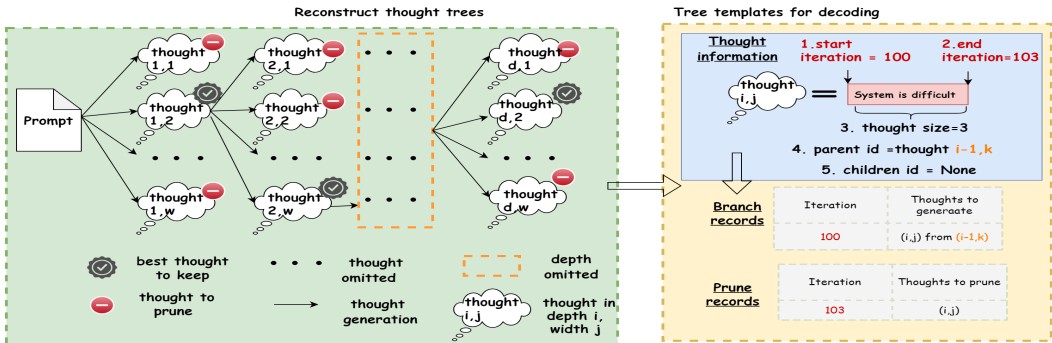

Figure 12: **The detailed procedure of reconstructing tree templates for multi-step reasoning**. (Left) Reconstructing reasoning trees from practical reasoning records as outlined in (Besta et al., 2023) involves capturing the following aspects: (1) the structure of trees, characterized by their depth $d$ and width $w$; (2) the token length associated with each thought; and (3) the best thought at each depth along with its corresponding score. For the task of document merging, the tree depth is set to $d = 3$, with a width of $w = 10$ at each depth. For sorting 128 numbers, the depth is reduced to $d = 10$, while maintaining the same width of $w = 10$. See details of tree topology for other multi-step reasoning tasks in Table 13. (Right) Utilizing the extracted thought information from Left, we can generate tree templates for decoding, encompassing *branch records* and *prune records*. These records are instrumental in guiding the tree decoding process to produce decoding trees that faithfully replicate the structure of the tree-of-thoughts.

- Multi-step reasoning (Yao et al., 2023; Hao et al., 2023; Besta et al., 2023): We recorded the tree shapes, prompts, and lengths of all thoughts from real reasoning task interactions (Besta et al., 2023), using these as guidance for tree decoding to validate DEFT's acceleration in thought generation of reasoning (the thought evaluation phase follows a similar pattern). See details of generation in Figure 12.
- Speculative decoding (Cai et al., 2024; Miao et al., 2023): We used the token tree topology from Medusa (Cai et al., 2024) and recorded real interaction data with APPS (Hendrycks et al., 2021) as prompt dataset, including the length of accepted tokens at each step. This served as guidance to simulate the bottleneck of speculative decoding—the attention computation during the token verification phase.

Table 13: **Details of generated workloads**. For multi-step reasoning, we include these 4 tasks from Besta et al. (2023): (1) Sorting 128 numbers (*sorting* in short); (2) Document merging (*document* in short); (3) Keyword counting (*keyword* in short); (4) Set intersection (*set* in short). $d$, and $w$ means depth and width of the tree, respectively. $t$ means the token tree size for speculative decoding, where the tree topology is from Medusa (Cai et al., 2024).

| Task | Tree Shape | Decoding Tree Source | Records Contents |
|---|---|---|---|
| Multi-step reasoning | *sorting*: $d = 10, w = 10$
*document*: $d = 3, w = 10$
*keyword*: $d = 5, w = 10$
*set*: $d = 8, w = 10$ | ToT-BFS in Besta et al. (2023) | Prompt (Besta et al., 2023), tree shape, thought size, branch records, prune records |
| Few-shot prompting | $d = 1, w = 10, 20, 30$ | – | – |
| Speculative decoding | $t = 32, 64, 128, 256$ | Medusa (Cai et al., 2024) | APPS (Hendrycks et al., 2021) Prompt, token tree shape, accepted token length per step |

**The rationality of our experiment paradigm.** Our experimental paradigm involves: first, obtaining decoding trees from real tree-based decoding tasks, and second, replicating these decoding trees exactly within the same framework by enforcing LLM inference, to investigate the impact of attention acceleration on wall clock time performance. This paradigm has two advantages:

- We can utilize decoding trees from real tasks as a benchmark within a unified system, enabling fair comparison of different attention algorithms in terms of decoding latency. This comparison is possible despite the algorithms being based on distinct systems, such as variations in memory management implementations for their kernels.

- We consider both the unique characteristics of tasks with diverse tree structures and the broader applicability of general tree-based decoding. See details of generated workloads for other multi-step reasoning tasks in Table 13.

## A.7    Additional Results

**Microbench of DEFT-Node for GPU utilization.**    We test the DEFT-Node and DEFT-Flatten on a speculative decoding with 64 queries and a prompt with 4k tokens. For DEFT-Node, the QKV partitioning is unbalanced as the node of the prompt is much longer than others (1 token for each query). As shown in Table 14, the metrics of GPU utilization are measured by NVIDIA Nsight Compute (NVIDIA, 2024). We can see DEFT-Flatten is better than DEFT-Node in both memory utilization (*Memory Throughput Ratio*) and calculation utilization (*Compute Throughput Ratio* and *Low Utilization Time Ratio*).

Table 14: **[GPU Utilization Microbenchmark] Latency of a single layer of Attention (in $\mu$s), SM Compute Throughput Ratio, Memory Throughput Ratio, and Low Utilization Ratio for DEFT on an NVIDIA A100 (80GB) using the LLama3-8B model (GQA).** The workload is speculative decoding with 64 queries and a prompt with 4k tokens. The *Compute Throughput Ratio* refers to the utilization of the Streaming Multiprocessors (SMs) in the GPU. The *Memory Throughput Ratio* represents the ratio between actual memory throughput and maximum bandwidth. The *Low Utilization Time Ratio* is defined as the proportion of time when the *Compute Throughput Ratio* falls below 5%.

| Method | Attention Latency | Compute Throughout Ratio | Memory Throughout Ratio | Low Utilization Time Ratio |
|---|---|---|---|---|
| DEFT-Node | 961.38 | 7.60% | 17.39% | 82.35% |
| DEFT-Flatten | 226.82 | 21.19% | 51.91% | 0.00% |

**Attention latency and IOs with breakdowns.**    The details of attention latency and IO comparison among DEFT and baselines are in Table 5 and Table 17, respectively. Note that the attention latency does not include the memory management overheads, which is the bottleneck for Tree Attention-Medusa. For example, it takes more than $80\%$ of end to end latency in speculative decoding with 32 queries.

Table 15: **Inference accuracy of DEFT in attention score and perplexity (PPL).** PPL is calculated after 400 iterations of decoding. Vanilla Attention is the implementation from Huggingface Transformers.

| | Relative Attention Error (↓) | | Perplexity (PPL) (↓) | | Relative PPL Error (↓) | |
|---|---|---|---|---|---|---|
| Attention Method | Attention Variations | | | | | |
| | MHA | GQA | MHA | GQA | MHA | GQA |
| Vanilla Attention | - | - | 1.000 | 1.002 | - | - |
| Radix Attention | 0.545% | 0.540% | 1.000 | 1.002 | $1 \times 10^{-6}$ | $1 \times 10^{-6}$ |
| DEFT-Node-Chunk | 0.403% | 0.403% | 1.000 | 1.002 | $1 \times 10^{-6}$ | $4 \times 10^{-6}$ |
| DEFT-Flatten | 0.407% | 0.404% | 1.000 | 1.002 | $1 \times 10^{-6}$ | $9 \times 10^{-7}$ |

**Inference accuracy of DEFT-Node-Chunk and DEFT-Flatten.**    In Section 3.1, equation 1 shows DEFT (including DEFT-Node, DEFT-Node-Chunk and DEFT-Flatten ) and vanilla attention are mathematically equivalent, which means DEFT is accurate. As demonstrated in Table 15, the DEFT attention scores may slightly differ (around $0.4\%$ relative error) compared to vanilla attention in Huggingface Transformers, but the generated tokens will hardly be different as well as PPL (1e-6 relative PPL error in the right part of Table 15). This discrepancy arises because floating-point operations on GPUs do not adhere to the associative law, even when two calculation processes are mathematically equivalent. Similar issues occur in other methods like radix attention and Flash-Decoding that introduce online Softmax and reduction as well, resulting in approximately $0.5\%$ relative attention score errors.

**Dynamic behaviors: per iteration latency.**    We visualize the per-iteration latency of DEFT-Node and DEFT-Flatten for a tree in the multi-step reasoning task–*sorting* in Figure 13, as the size and topology of the decoding tree change with each iteration. This comparison highlights the sensitivity of these two split strategies to changes in tree size. We observe a strong positive correlation between the ratio of per-iteration latency of DEFT-Node and DEFT-Flatten (Speedup Ratio) and the dispersion of tree node sizes. This correlation arises because the performance of DEFT-Flatten remains relatively stable, whereas the performance of DEFT-Node is more strongly influenced by the topology of the tree. DEFT-Flatten provides a stable speedup of approximately $1.75\times$ compared to DEFT-Node.

Table 16: **Average attention latency (in seconds) for tree-based decoding and its impact on decoding latency.** Here, $b$ represents the tree width, and $t$ denotes the token tree size (i.e., the number of tree-structured queries). *Attention Speedup over the best attention* refers to the speedup of DEFT-Flatten compared to the best baseline (typically *Tree Attention-Medusa*) in attention calculation. *Radix Attention* is the best baseline for decoding latency. Note that KV cache management is not included in the attention latency. ⋆ denotes out-of-memory (OOM) errors for the A100 80GB GPU. For more details on decoding latency, see Table 5.

| Memory | Method | Few-shot Prompting | | | Multi-Step Reasoning | | | | Speculative Decoding | | | |
|---|---|---|---|---|---|---|---|---|---|---|---|---|
| | | b=20 | b=30 | b=50 | *Sorting* | *Document* | *Keyword* | *Set* | t=32 | t=64 | t=128 | t=256 |
| Unpaged | Flash-Decoding | 43.49 | 66.10 | 110.09 | 160.67 | 105.80 | 12.14 | 19.96 | 340.09 | 692.88 | ⋆ | ⋆ |
| | Tree Attention-Medusa | 3.93 | 7.51 | 9.57 | 38.64 | 29.10 | 2.62 | 3.96 | 22.40 | 26.31 | 41.10 | 68.28 |
| Paged | Radix Attention | 5.99 | 7.30 | 9.96 | 39.37 | 24.69 | 3.11 | 5.13 | 25.73 | 40.47 | 76.10 | 145.43 |
| | DEFT-Flatten . | 3.47 | 4.07 | 5.87 | 28.41 | 21.45 | 2.57 | 3.83 | 13.15 | 16.79 | 24.46 | 40.56 |
| | Attention Speedup over the best attention. | 1.13× | 1.63× | 1.70× | 1.36× | 1.15× | 1.02× | 1.03× | 1.70× | 1.57× | 1.68× | 1.68× |
| | *Attention Speedup over Radix Attention* | 1.73× | 1.63× | 1.70× | 1.39× | 1.15× | 1.21× | 1.34× | 1.96× | 2.41× | 3.11× | 3.59× |
| | *Decoding Speedup over Radix Attention* | 1.24× | 1.28× | 1.33× | 1.10× | 1.03× | 1.03× | 1.05× | 1.29× | 1.50× | 1.91× | 2.23× |

Table 17: **Average end-to-end IO (TB) during decoding.** Data format is Left/Right: *(Left)* KV Cache IO; *(Right)* partial results IO, including $QK^T$, $QK^\top/s_c$, Mask $M$, $M + QK^\top/s_c$ and Softmax. $b$ means tree width. $t$ denotes the token tree size (i.e., the number of tree-structured queries). ⋆ means out of memory for A100 80GB.

| Method | Few-shot Prompting | | | Multi-Step Reasoning | | | | Speculative Decoding | | | |
|---|---|---|---|---|---|---|---|---|---|---|---|
| | b=20 | b=30 | b=50 | *Sorting* | *Document* | *Keyword* | *Set* | t=32 | t=64 | t=128 | t=256 |
| Flash-Decoding | 17.62/0.00 | 26.43/0.00 | 44.05/0.00 | 59.96/0.00 | 39.74/0.00 | 4.68/0.00 | 7.01/0.00 | 128.72/0.00 | 255.16/0.00 | ⋆ | ⋆ |
| Tree Attention-Medusa | 1.68/1.05 | 2.10/1.98 | 2.94/4.61 | 12.40/3.69 | 10.57/3.24 | 0.58/0.18 | 1.04/0.27 | 4.02/4.03 | 4.15/8.33 | 4.18/16.77 | 4.32/34.70 |
| Radix Attention | 17.62/0.00 | 26.43/0.00 | 44.05/0.00 | 59.96/0.00 | 39.74/0.00 | 4.68/0.00 | 7.01/0.00 | 131.45/0.00 | 256.79/0.00 | 522.05/0.00 | 1044.10/0.00 |
| DEFT-Flatten | 1.68/0.00 | 2.10/0.00 | 2.94/0.00 | 12.40/0.01 | 10.57/0.01 | 0.58/0.00 | 1.04/0.00 | 4.10/0.00 | 4.11/0.00 | 4.16/0.00 | 4.35/0.00 |
| IO reduction of DEFT-Flatten(%) | 90.47/100.00 | 92.1/100.00 | 93.33/100.00 | 79.32/99.73 | 73.40/99.70 | 87.61/100.00 | 85.16/100.00 | 96.88/100.00 | 98.40/100.00 | 99.20100.00 | 99.58/100.00 |

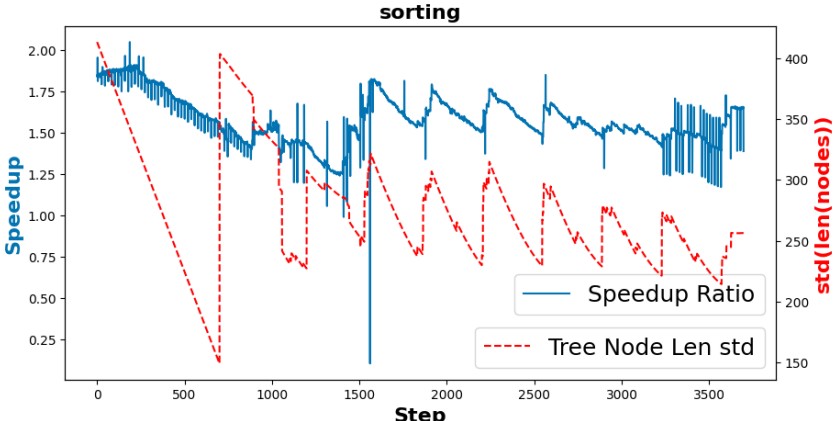

Figure 13: Comparison of split strategies DEFT-Node and DEFT-Flatten in *sorting* task. *Speedup ratio* refers to the ratio between the per iteration latency of DEFT-Node and DEFT-Flatten. *Tree Node Len std* represents the standard deviation of the tree node lengths for each iteration.

**Ablation: The influence of width in decoding trees.** We observe that the effectiveness of attention speedup varies with different decoding tree topologies. Considering the simplest tree structure, a prompt with several suffixes—given a prompt that is not very short, one of the most important factors for speedup is the extent to which we can reuse its KV cache IO. This can be measured by the width of the tree. More specifically, it is determined by the number of queries per iteration. Therefore, we fix the prompt length at 4000 and vary the width of the decoding tree in few-shot prompting (which also indicates how many requests share the same prompt). Then, as shown in Figure 14, we evaluate DEFT-Flatten with the best baseline in attention calculation– Tree Attention-Medusa (Cai et al., 2024) (Medusa-Attn in the figure), as well as the best baseline in decoding latency– Radix Attention (Zheng et al., 2023), for the per-iteration latency over time. We have the following observations:

1. When the tree width is 10, the attention overhead of DEFT-Flatten is nearly the same as Tree Attention-Medusa because the IO overhead of the dense causal mask (DCM) is small compared to

that of the KV cache, but it is still $2\times$ faster in attention latency than Radix Attention thanks to the KV IO reuse.

2. As the tree width increases, the attention computation overhead of Tree Attention-Medusa grows faster because the size of the DCM is directly related to the tree width. A larger tree width means the IO of the DCM grows rapidly.

3. Since the tree topology consists of a fixed prefix with several suffixes, a larger tree width allows the prompt prefix's KV cache to be reused more frequently during IO. This leads to a more significant decoding speedup—$1.24\times$ with a width of $w = 20$, and $1.33\times$ with a width of $w = 50$—compared to Radix Attention.

4. As iterations progress, the length of the suffixes gradually approaches the length of the prefix, leading to a decrease in the speedup of DEFT-Flatten compared with Radix Attention.

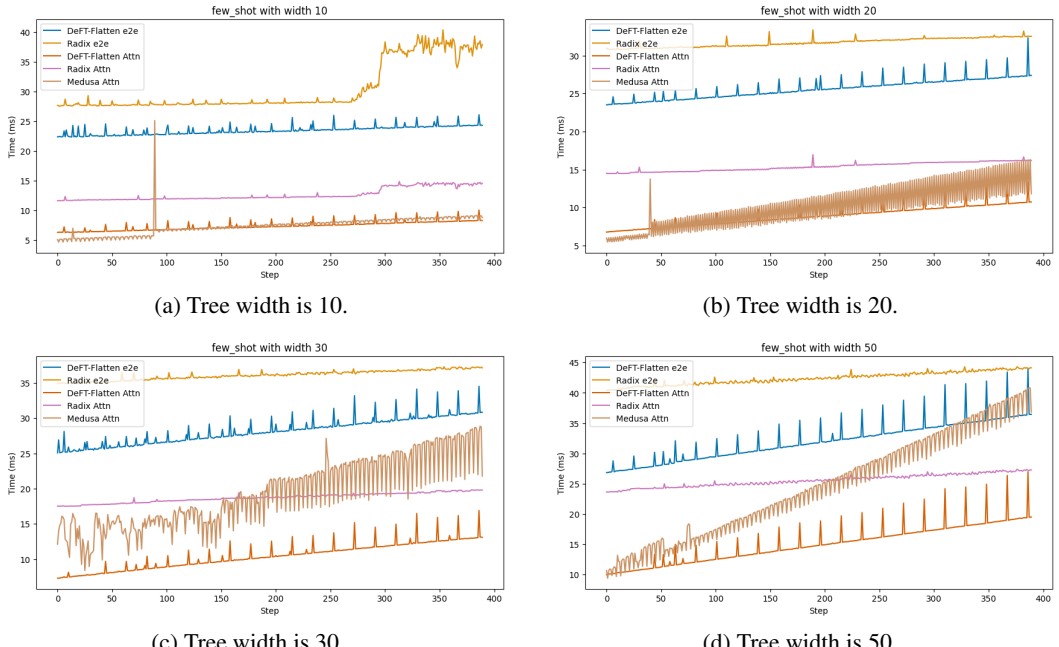

(a) Tree width is 10.

(b) Tree width is 20.

(c) Tree width is 30..

(d) Tree width is 50.

Figure 14: Per iteration latency for few-shot prompting tasks with different tree width. *e2e* means decoding latency(optimal end-to-end latency), while *Attn* means only the attention overhead.

**Ablation: The influence of chunk size in KV splitting.** In the implementation of DEFT-Flatten and DEFT-Node-Chunk, we selected a regular size (128) in General matrix multiply (GEMM) as the block/chunk size of KV cache during the attention calculation. We added an ablation study of chunk size influence on speedup, as shown in Figure 15. The chunk size selection is a trade-off between IO redundancy and threadblock scheduling: a larger chunk size means less redundancy of Query IO but may cause potential idle SMs of GPUs due to fewer threadblocks during GPU scheduling. Conclusions of Figure 15: (1) The best chunk size is influenced by both sequence length and query numbers (batch size); (2) DeFT-Flatten can outperform DeFT-Node-Chunk in all chunk sizes we test.

**Ablation: The influence of prompt length.** We set the prompt length from 1k to 20k tokens, and the generation length is set to 1k tokens, to simulate a long prompt short generation case. Our workloads are speculative decoding with token tree size of 32 and 64. The ablation is based on Llama3.1-8B model and NVIDIA A100 80GB. We have three metrics: time per output token (Figure 16), decoding latency (Figure 17) and attention latency (Figure 18).

**Ablation: The influence of both model size and prompt length.** In Table 6 and Table 8 of Section 4.4, we show the influence of prompt length and model size, individually. Here, we present the ablation study of both model size and prompt length for *sorting* task, as shown in Table 18. With a longer prompt, DeFT shows more pronounced speedup in the same model, since the attention overhead is proportional to the token count in the decoding tree, while the FFN overhead remains nearly constant for the same model. For a fixed prompt length, with the larger model—Codellama-34B, DEFT-Flatten achieves slightly reduced but still significant (up to 1.28x) decoding speedup.

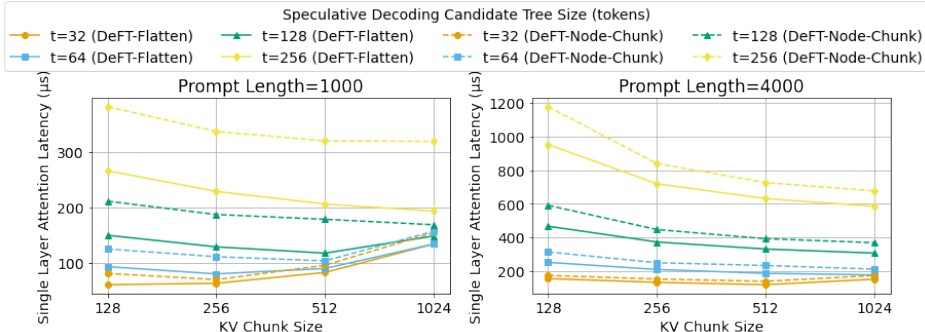

Figure 15: **Ablation study for KV chunk size with DEFT**. $t$ is the token tree size in speculative decoding.

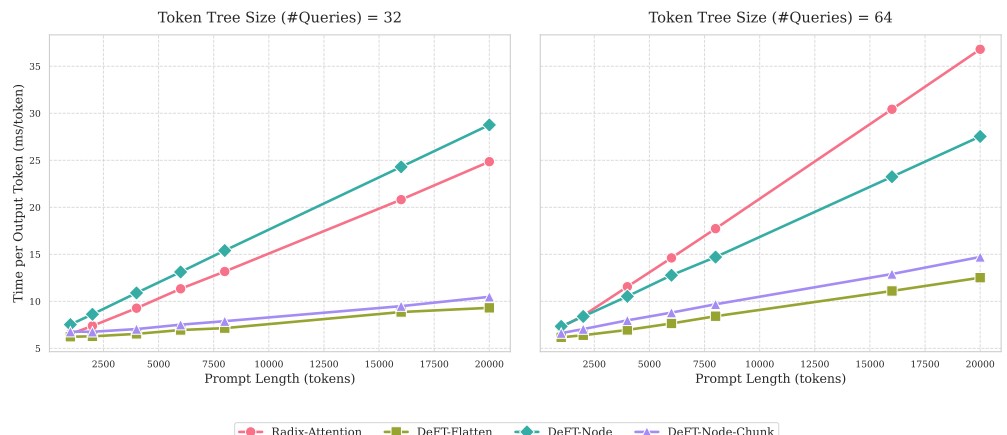

Figure 16: Time per output token(TPOT) of DEFT with different prompt lengths in speculative decoding.

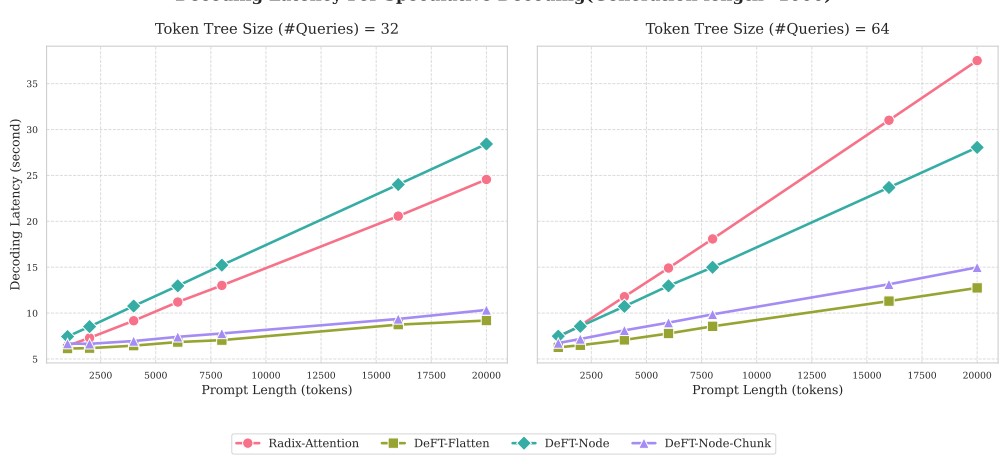

Figure 17: Decoding latency of DEFT with different prompt lengths in speculative decoding.

The performance reduction is attributed to a lower A/F-LR, as the FFN overhead is greater in larger models.

**Ablation: Different GPUs.** See Table 19. DEFT-Flatten can have obvious speedup in RTX 4090 as well because the memory hierarchy of GPUs is nearly the same— large but slow global memory and small but fast shared memory.

Figure 18: Attention latency of DEFT with different prompt lengths in speculative decoding.

Table 18: **[Ablation Study of Model Size and Prompt Length] Comparison of decoding speedup and Attention/FFN latency ratio (A/F-LR) between DEFT and Radix Attention for Codellama-34B and Codellama-7B across varying prompt lengths in the `sorting` reasoning task.** Radix Attention is the best baseline in decoding latency.

| Metric | Model Size | Prompt Length=1k | Prompt Length=5k | Prompt Length=8k |
|---|---|---|---|---|
| Decoding latency Speedup | 7B | $1.09\times$ | $1.37\times$ | $1.53\times$ |
| | 34B | $1.03\times$ | $1.18\times$ | $1.28\times$ |
| Radix Attention's A/F-LR | 7B | 1.12 | 1.89 | 2.50 |
| | 34B | 0.48 | 0.86 | 1.16 |
| DEFT-Flatten's A/F-LR | 7B | 0.89 | 1.09 | 1.25 |
| | 34B | 0.42 | 0.57 | 0.67 |

Table 19: **[Different GPUs] Speedup of DEFT in average attention latency (second) with NVIDIA RTX 4090 (24GB) for LLama3-8B model(GQA).** Radix Attention is the best baseline in decoding latency.

| Memory | Method | Few-shot Prompting-b=30 | Multi-Step Reasoning-*Sorting* | Speculative Decoding-t=64 |
|---|---|---|---|---|
| | Radix Attention | 4.26 | 26.36 | 33.63 |
| Paged | DEFT-Node-Chunk | 3.07 | 24.61 | 15.39 |
| | DEFT-Flatten | 2.95 | 23.86 | 14.04 |
| | *Attention Speedup over the best decoding* | $1.44\times$ | $1.10\times$ | $2.40\times$ |

**Ablation: Different Model Architectures.** See Table 20 and Table 21. DEFT-Flatten can both accelerate the attention computation of LLM models with different architectures (MHA and GQA) significantly.

Table 20: **[Different Model Architectures(GQA)] Speedup of DEFT in average attention latency (second) with NVIDIA A100(80GB) for Codellama-34B model(GQA).** Radix Attention is the best baseline in decoding latency.

| Memory | Method | Few-shot Prompting-b=30 | Multi-Step Reasoning-*Sorting* | Speculative Decoding-t=64 |
|---|---|---|---|---|
| | Radix Attention | 16.85 | 95.14 | 164.33 |
| Paged | DEFT-Node-Chunk | 16.15 | 103.15 | 81.74 |
| | DEFT-Flatten | 9.62 | 84.30 | 48.76 |
| | *Attention Speedup over the best decoding* | $1.75\times$ | $1.13\times$ | $3.37\times$ |

## A.8 DEFT-NODE ALGORITHM

Table 21: **[Different Model Architectures(MHA)] Speedup of DEFT in average attention latency (second) with NVIDIA A100(80GB) for Codellama-7B model(MHA).** Radix Attention is the best baseline in decoding latency.

| Memory | Method | Few-shot Prompting-b=30 | Multi-Step Reasoning-*Sorting* | Speculative Decoding-t=64 |
|---|---|---|---|---|
| | Radix Attention | 12.39 | 53.96 | 96.55 |
| Paged | DEFT-Node-Chunk | 10.12 | 54.20 | 48.96 |
| | DEFT-Flatten | 8.24 | 43.91 | 36.48 |
| | *Attention Speedup over the best decoding* | 1.50× | 1.23× | 2.65× |

---

**Algorithm 1** DEFT-Node Algorithm-Phase 1: QKV Preparation.

---

**Input:** query $Q \in R^{(b_q,d)}$, Key cache list $KL = (K_0,...K_{N-1})$, Value cache list $VL = (V_0,...V_{N-1})$ for each sequence node in the tree, where $N$ is the total number of sequences in a tree, and Tree $T$ with its topology information.
**for** each $q$ in $Q$ with its global index $idx$ **do**
    /*Get KV indices of all prefixes' for a query.*/
    $QMapKV[idx]$=GetPrefixKVIndices($q, KL, VL, T$)
**end for**
**for** each seq's KV cache $K_i, V_i$ in $KL, VL$ with its KV indice $i$ **do**
    /*Group each sequence's KV with all queries that share it.*/
    $Q_i$= GroupQueryToKV($Q, K_i, V_i, T$) $\in R^{b_i,d} \subset Q$
    $KVMapQ[i] = Q_i$
**end for**
**Return** QMapKV, KVMapQ

---

DEFT-Node has two phases-**Phase 1-QKV Preparation** and **Phase 2-Attention Calculation**.
**Phase 2-Attention Calculation** of DEFT has two stages.
1. **Stage 1: Calculate Partial Attentions.** We will apply Flash Attention of all QKV groups obtained after **Phase 1-QKV Preparation** of DEFT, to get partial attention and LogSumExp.
2. **Stage 2: Global Reduction.** We will remap partial attention and LogSumExp based on each query, and get final attention based on global reduction similar to Flash-Decoding (Dao et al., 2023).

---

**Algorithm 2** DEFT-Node Algorithm-Phase 2: Attention Calculation.

---

**Input:** query $Q \in R^{(b_q,d)}$, Key cache list $KL = (K_0, ...K_{N-1})$, Value cache list $VL = (V_0, ...V_{N-1})$ for each sequence node in the tree, where $N$ is the total number of sequences in a tree, and Tree $T$ with its topology information. QKV group information $QMapKV, KVMapQ$ from **QKV Preparation Phase**.

**for** each $q$ in $Q$ with its global index $idx$ **do**

   /*Allocate to store LogSumExp of $Q@K^T$ grouped by query.*/

   $LogSumExp[idx] = \{\}$

   /*Allocate to store partial results of $SoftMax(Q@K^T)V$ for each query.*/

   $O[idx] = \{\}$

**end for**

/*Allocate space for output after reduction.*/

$FO = (0)_{b_q \times d} \in R^{(b_q,d)}$

**for** each seq's KV cache $K_i, V_i \in R^{(b_{kv},d)}, R^{(b_{kv},d)}$ in $KL, VL$ with its KV indice $i$ **do**

   **# Unroll for loop to SMs**

   $Q_i = KVMapQ[i] \in R^{(b_i,d)}$

   /*Get partial attention $o_i$ for each QKV group, LogSumExp $lse_i$ of $Q@K^T$ in row for reduction.*/

   $o_i, lse_i$ = FlashAttention$(Q_i, K_i, V_i)$

   $\in R^{(b_i,d)}, R^{b_i}$

   /*Map the partial results back to each query for reduction.*/

   **for** each query $q$ in $Q_i$ with its group index $gp\_idx$ and global index $idx$ in $Q$ **do**

     **if** $i \in QMapKV[idx]$ **then**

       $LogSumExp[idx].append(lse_i[gp\_idx])$

     **end if**

   **end for**

**end for**

**for** each $q$ in $Q$ with its global index $idx$ **do**

   **# Unroll for loop to SMs**

   **if** len($O[idx]$)==len($QMapKV[idx]$) **then**

     /*Global reduction after collecting all partial results from QKV groups that contains $q$.*/

     $LSE_{cat}$ = CatTensor($LogSumExp[idx]$)

     $LSE_{max}$ = RowMax($LSE_{cat}$)

     $Mid\_L = 0, Mid\_O = 0^{(1,d)}$

     **for** each $lse_j$ in $LogSumExp[idx]$ **do**

       $new\_exp = e^{lse_j - LSE_{max}}$

       $Mid\_L = Mid\_L + new\_exp$

     **end for**

     **for** each $lse_j, o_j$ in $LogSumExp[idx], O[idx]$ **do**

       $new\_exp = e^{lse_j - LSE_{max}}$

       $Mid\_O = Mid\_O + new\_exp@o_j / Mid\_L$

     **end for**

     $FO[idx] = Mid\_O$

   **end if**

**end for**

**Return** $FO$

---

## A.9 DEFT-FLATTEN ALGORITHM

The algorithm (noted as DEFT-Node) in Appendix A.8 adopts a node-granularity split strategy, which is quite simple. However, when the token lengths of different nodes in a decoding tree are very unbalanced, it might introduce inefficient calculation due to the unbalanced workload in on-chip SMs of GPUs.

Therefore, we can split the decoding tree in a more balanced way– in subtree-granularity. We show the DEFT-Flatten algorithm as follows, which also consists of two stages similar to DEFT-Node.

---

**Algorithm 3** DEFT-Flatten Algorithm-Phase 1: QKV Preparation.

---

**Input:** query $Q \in R^{(b_q,d)}$, Key cache list $KL = (K_0, ...K_{N-1})$, Value cache list $VL = (V_0, ...V_{N-1})$ for each sequence node in the tree, where $N$ is the total number of sequences in a tree, and Tree $T$ with its topology information. Subtree size $S_t$, which means each subtree after tiling contains at most $S_t$ tokens.

/*Evenly slice/blockwise the Tree KV cache (with $n_T$ tokens in the tree ) to subtrees.*/

SubInfo, KSub, VSub =Slice( KL, VL, $S_t$, $T$)

/*Notes: (1) subtree number $m = Ceil(n_T/S_t)$;

(2) subtrees' KV cache $KSub = (Kb_0, ..., Kb_{m-1})$, $VSub = (Vb_0, ..., Vb_{m-1})$;

(3) subtree information $SubInfo = (Sb_0, ..., Sb_{m-1})$, where each subtree i has $Sb_i = (ofs_0, ...ofs_{n_{b_i}-1})$ to record the offset of each node in the subtree KV cache, with $n_{b_i}$ as the total number of nodes in subtree $i$. */

**for** each subtree's KV cache $Kb_i, Vb_i$ in $KSub, VSub$ with its subtree ID $i$ **do**

    /*Group each subtree's KV with all queries that share it.*/

    $Q_i$= GroupQueryToKV($Q, Kb_i, Vb_i, T$) $\in R^{b_i,d} \subset Q$

    $KVMapQ[i] = Q_i$

    **for** each query $q$ in $Q_i$ with a global index $idx$ in $Q$ **do**

        $QMapKV[idx].append(i)$

    **end for**

    /*Add a causal mask as different nodes in a subtree could be shared by different queries.*/

    $CausalMask[i] = GetBitMask(Q_i, Kb_i, Vb_i, T)=(CM_0, ...CM_{n_{b_i}-1})$

    where $n_{b_i}$ is the total number of nodes in the subtree, and $CM_i$ is a 64-bit int bit mask for node i.

    /*E.g, 100....00 **with 1 in bit 0, means the** $Q_i[0]$ **does not share KV cache of node i in the subtree.**/

**end for**

**Return** QMapKV, KVMapQ, CausalMask,SubInfo

---

---

**Algorithm 4** DEFT-Flatten Algorithm-Phase 2: Attention Calculation.

---

**Input:** query $Q \in R^{(b_q,d)}$, Key cache list in subtree-granularity KSub=$(Kb_0,...,Kb_{m-1})$, Value cache list in subtree VSub = $(Vb_0,...,Vb_{m-1}$ for $m$ subtrees after tiling based on Tree $T$ with its topology information. QKV group information $QMapKV$, $KVMapQ$, causal mask $CausalMask$ and subtree information $SubInfo$ from **QKV Preparation Phase**.

**for** each $q$ in $Q$ with its global index $idx$ **do**
  /\***Allocate to store LogSumExp of** $Q@K^T$ **grouped by query.**\*/
  $LogSumExp[idx] = \{\}$
  /\***Allocate to store partial results of** $SoftMax(Q@K^T)V$ **for each query.**\*/
  $O[idx] = \{\}$
**end for**
/\***Allocate space for output after reduction.**\*/
$FO = (0)_{b_q \times d} \in R^{(b_q,d)}$
**for** each subtree's KV cache $Kb_i, Vb_i \in R^{(b_{kv},d)}, R^{(b_{kv},d)}$ in $KSub, VSub$ with subtree ID $i$ **do**
  # Unroll for loop to SMs
  $Q_i = KVMapQ[i] \in R^{(b_i,d)}$
  /\***Reconstruct mask for attention calculation based on** $CausalMask$ **and** $SubInfo$\*/
  $bitmask = CausalMask[i] \in R^{n_{b_i}}$,where $n_{b_i}$ is the total number of nodes for subtree i.
  $SubOfst = SubInfo[i] \in R^{n_{b_i}}$
  $mask = ReconstructMask(bitmask, SubOfst) \in R^{(b_i,b_{kv})}$
  /\***Get partial attention** $o_i$ **for each QKV group, LogSumExp** $lse_i$ **of** $Q@K^T$ **in row for reduction.**\*/
  $o_i, lse_i = \text{FlashAttention}(Q_i, Kb_i, Vb_i, mask)$
  $\in R^{(b_i,d)}, R^{b_i}$
  /\***Map the partial results back to each query for reduction.**\*/
  **for** each query $q$ in $Q_i$ with its group index $gp\_idx$ and global index $idx$ in $Q$ **do**
    **if** $i \in QMapKV[idx]$ **then**
      $LogSumExp[idx].append(lse_i[gp\_idx])$
    **end if**
  **end for**
**end for**
**for** each $q$ in $Q$ with its global index $idx$ **do**
  # Unroll for loop to SMs
  **if** len($O[idx]$)==len($QMapKV[idx]$) **then**
    /\***Global reduction after collecting all partial results from QKV groups that contains** $q$.\*/
    $LSE_{cat} = \text{CatTensor}(LogSumExp[idx])$
    $LSE_{max} = \text{RowMax}(LSE_{cat})$
    $Mid\_L = 0, Mid\_O = 0^{(1,d)}$
    **for** each $lse_j$ in $LogSumExp[idx]$ **do**
      $new\_exp = e^{lse_j - LSE_{max}}$
      $Mid\_L = Mid\_L + new\_exp$
    **end for**
    **for** each $lse_j, o_j$ in $LogSumExp[idx], O[idx]$ **do**
      $new\_exp = e^{lse_j - LSE_{max}}$
      $Mid\_O = Mid\_O + new\_exp@o_j / Mid\_L$
    **end for**
    $FO[idx] = Mid\_O$
  **end if**
**end for**
**Return** $FO$

---

