# OpenReview forum: "DeFT: Decoding with Flash Tree-attention for Efficient Tree-structured LLM Inference"
_ICLR.cc/2025/Conference — ICLR 2025 Spotlight_

### Official Review · Reviewer_6baS · 2024-10-25

**Soundness:** 4
**Presentation:** 3
**Contribution:** 3
**Rating:** 8
**Confidence:** 4

**Summary:**

The paper presents DEFT (Decoding with Flash Tree-Attention), a hardware-efficient algorithm that optimizes large language model (LLM) inference for tree-based decoding tasks like few-shot prompting and multi-step reasoning. Current systems struggle with redundant Key-Value (KV) cache loading and poor load balancing, causing inefficient memory use and low GPU utilization. DEFT solves this with KV-Guided Grouping, which reuses shared prefixes to reduce KV cache access, and Flattened Tree KV Splitting, which improves GPU efficiency. Implemented with OpenAI Triton, DEFT achieves significant speedups in attention latency compared to existing methods

**Strengths:**

Relevance: The paper tackles a timely and significant problem in optimizing LLM inference for tree-based decoding applications, which is highly relevant to current AI research and deployment.
Originality: Introduces a novel attention algorithm, DEFT, leveraging KV-Guided Grouping and Flattened Tree KV Splitting to address memory access inefficiencies and load balancing.
Theoretical Justification: Provides solid theoretical analysis to justify the proposed methods, including IO complexity analysis and discussions on the correctness of the algorithm.
Empirical Validation: Demonstrates significant improvements in end-to-end latency and attention computation across multiple tasks (few-shot prompting, multi-step reasoning, speculative decoding) compared to state-of-the-art baselines. The supplementary material includes extensive experimental results and ablation studies, strengthening the empirical validation.
Comparison with Concurrent Works: The supplementary material provides detailed comparisons with concurrent works, clarifying the advantages of DEFT in handling multi-level tree decoding and addressing unbalanced workloads.
Scalability: The authors provide results demonstrating DEFT's scalability to larger models (up to 34B parameters) and different hardware setups.
Accuracy Preservation: The paper includes analysis showing that DEFT maintains model accuracy, with negligible differences in attention scores and perplexity compared to baseline methods

**Weaknesses:**

Presentation Clarity: While the supplementary material improves clarity, some sections of the main paper remain dense, and the inclusion of key explanations from the supplementary material into the main text could further enhance understanding. Significant critical information is gained through the supplementary material, specifically regarding reproducibility and algorithm details, which would benefit from inclusion in the main text.
Limited Discussion on Energy Efficiency: The paper still focuses primarily on speedup metrics, and while memory access reduction implies energy efficiency, an explicit discussion or measurement of energy consumption would strengthen the work.
Applicability in Varying Scenarios: Although the authors include experiments with varying tree widths and prompt lengths, further exploration of scenarios with minimal shared prefixes or very small tree widths would provide a more comprehensive understanding of DEFT's applicability.

**Questions:**

Integration of Supplementary Material: Could the authors consider integrating key explanations and findings from the supplementary material into the main paper to improve readability and clarity for readers who may not delve into the appendix?
Energy Efficiency Metrics: While DEFT reduces IO operations, have the authors considered measuring the impact on energy consumption or providing an analysis of energy efficiency improvements?
Minimal Shared Prefixes Scenarios: How does DEFT perform in scenarios where the shared prefix is minimal or the tree width is very small? Are there any overheads introduced in such cases compared to existing methods?
Realistic Scalability: Do the authors foresee any limitations or challenges in extending DEFT to more common larger model sizes (e.g., 70B parameters, or 400B) or to different model architectures beyond those tested? These larger models generally excel at complex multi-step reasoning tasks compared to the <32B counterparts, which may reveal different patterns in inference and could affect the effectiveness or accuracy retention of your approach.

**Details Of Ethics Concerns:**

The paper makes a significant contribution to optimizing LLM inference for tree-based decoding tasks, introducing novel methods that are both theoretically sound and empirically validated. The authors have addressed previous concerns through additional material, improving the clarity and robustness of the work. Therefore, I recommend acceptance

---

> ### Author Response · Authors · 2024-11-22
> **Response to Reviewer 6bas(1/2)**
>
> Thank you for your helpful feedback and insightful questions.
>
> **Weaknesses:**
>
> > W1: Presentation Clarity: While the supplementary material improves clarity, some sections of the main paper remain dense, and the inclusion of key explanations from the supplementary material into the main text could further enhance understanding. Significant critical information is gained through the supplementary material, **specifically regarding reproducibility and algorithm details**, which would benefit from inclusion in the main text.
> >
>
> Thanks for your suggestion for improving the clarity of our paper! See CQ3.
>
> > W2: Limited Discussion on Energy Efficiency: The paper still focuses primarily on speedup metrics, and while memory access reduction implies energy efficiency, an explicit discussion or measurement of energy consumption **would strengthen the work**.
> >
>
> We agree that energy efficiency is an important metric for real-world deployment.  See Q2.
>
> > W3: Applicability in Varying Scenarios: Although the authors include experiments with varying tree widths and prompt lengths, further exploration of scenarios with **minimal shared prefixes** or very small tree widths would provide a more comprehensive understanding of DEFT's applicability.
> >
>
> See Q3.
>
> **Questions:**
>
> > Q1: Integration of Supplementary Material: Could the authors consider integrating key explanations and findings from the supplementary material into the main paper to improve readability and clarity for readers who may not delve into the appendix?
> >
>
> Thanks for your suggestion! We summarize the improvement of writing in CQ3.
>
> > Q2: Energy Efficiency Metrics: While DEFT reduces IO operations, have the authors considered measuring the impact on energy consumption or providing an analysis of energy efficiency improvements?
> >
>
> Thanks for pointing out a potential advantage of DeFT in energy efficiency!
>
> The main focus of this work is latency and our roadmap is to add energy efficiency after we expand the DeFT to multi-GPU versions. We would like to explore the user-sensitive metrics like latency first then explore the service side metrics like energy efficiency. But we do believe energy is indeed a potential advantage of DeFT as the calculation is nearly the same but the memory access is much less—whether the memory access takes the most of energy cost still requires future experiments to verify.
>
> > Q3: Minimal Shared Prefixes Scenarios: How does DEFT perform in scenarios where the shared prefix is minimal or the tree width is very small? Are there any overheads introduced in such cases compared to existing methods?
> >
> - For the scenarios when the shared prefix is small,  DeFT will have less speedup, because it will degenerate to Radix Attention. When tree width = 10, the shared prompt length reaches 1k, and there will be 1.39X attention speedup and 1.09X wall clock speedup, as shown in Table 7 of our paper.
> - For the scenarios when treewidth is small, we compare DeFT-Flatten with the SOTA Radix Attention with the settings of tree size T=5 and 10 in speculative decoding, and treewidth=1,2,5 in few shot prompting. The model is llama3-8B and the GPU is A100 80GB. The results are as follows.  We can see the speedup of attention latency is still obvious(up to 2.20x) but the end-to-end latency is up to 1.21x. The reason is that when the total number of tokens in a decoding tree is small,  the bottleneck is the FFN rather than attention.
> - Table: Speculative decoding (t=5/10,  other settings are the same as our paper including prompt length=~1k tokens).
>
> | token tree size T | method | end-to-end latency (s) | attention latency (s) | attention speedup | e2e speedup |
> | --- | --- | --- | --- | --- | --- |
> | 5 | DeFT-Flatten | 44.34 | 4.15 | 1.73X | 1.18X |
> | 5 | Radix Attention | 52.38 | 7.19 | - | - |
> | 10 | DeFT-Flatten | 45.55 | 4.65 | 2.20X | 1.21X |
> | 10 | Radix Attention | 55.44 | 10.25 | - | - |

---

> ### Author Response · Authors · 2024-11-22
> **Response to Reviewer 6bas(2/2)**
>
> **(continued for Q3)**
> - Table: Few shot prompting (treewidth = 1,2,5, other settings are the same as our paper including prompt length=~4k tokens).
>     - To point out, the attention latency in treewidth 1 is just because of the implementation difference between DeFT-Flatten and Radix Attention.
>     - We can see when we expand the treewidth from 1 to 5, the attention latency in DeFT-Flatten increases by only 0.24s, while it increases by 0.39s due to a lack of awareness for reusing KV cache memory access in Radix Attention.
>
> | treewidth | method | end-to-end latency(s) | attention latency(s) | attention speedup | e2e speedup |
> | --- | --- | --- | --- | --- | --- |
> | 5 | DeFT-Flatten |  9.57  | 2.67  | 1.58X | 1.16X |
> | 5 | Radix Attention |  11.07 | 4.21 | - | - |
> | 2 | DeFT-Flatten | 9.40 | 2.48 | 1.62X | 1.15X |
> | 2 | Radix Attention | 10.83 | 4.02 | - | - |
> | 1 | DeFT-Flatten | 9.34 | 2.43 | 1.57X | 1.14X |
> | 1 | Radix Attention | 10.64 | 3.82 | - | - |
> - Compared with Radix Attention, DeFT-Flatten just introduced a little bit more (<5% of end-to-end latency) overhead in QKV preparation for serializing the memory addresses and grouping, yielding more than 1.57X/1.14X speedup in attention/end-to-end latency on the few shot prompting workloads with treewidth 1-5 shown in the above table.  Larger treewidth would have a much more obvious speedup while the QKV preparation cost is still negligible.
>
> > Q4: Realistic Scalability: Do the authors foresee any limitations or challenges in extending DEFT to more common larger model sizes (e.g., 70B parameters, or 400B) or to different model architectures beyond those tested? These larger models generally excel at complex multi-step reasoning tasks compared to the <32B counterparts, which may reveal different patterns in inference and could affect the effectiveness or accuracy retention of your approach.
> >
>
> We would like to point out DeFT is an accurate attention algorithm, which means it would not bring a difference in accuracy or effectiveness, as shown in Table 15, Appendix A7. The major challenges in extending DeFT to larger model sizes lie in efficiency:
>
> - [distributed setting needed for long-context/large-treewidth reasoning with large models] A larger model size takes larger memory storage, which means the memory remaining for KV cache storage would be less. We need to expand DeFT to a multi-GPU setting for long-context or large treewidth scenarios.
>     - It can be implemented by tensor parallelism easily as TP is orthogonal to DeFT;
>     - If we want additional sequence parallelism, it would be complex in system designs as discussed in CQ1.
> - [distributed setting brings more challenges] A distributed setting brings an additional cost of communication among different GPUs.  We have these open questions to answer in the future:
>     - How to overlap this cost with the computation for overhead hidden?
>     - How to schedule the requests of different decoding trees in multiple GPUs for large throughput without sacrificing the latency too much?

---

### Official Review · Reviewer_2d42 · 2024-11-02

**Soundness:** 3
**Presentation:** 3
**Contribution:** 3
**Rating:** 8
**Confidence:** 4

**Summary:**

The paper presents DEFT (Decoding with Flash Tree-Attention), an optimized algorithm for efficient inference in tree-structured large language model (LLM) tasks, such as few-shot prompting, multi-step reasoning, and speculative decoding. Existing methods face inefficiencies from redundant memory access of shared prefixes and unbalanced GPU workload distribution, which leads to low utilization and slower processing. DEFT addresses these issues with two key techniques: KV-Guided Grouping, which minimizes memory access by loading shared prefix data only once, and Flattened Tree KV Splitting, which enhances GPU utilization by evenly distributing workload across GPU units. Implemented on Triton, DEFT achieves up to 2.5x faster end-to-end speeds by significantly reducing memory operations, making it highly effective for complex, tree-based LLM applications.

**Strengths:**

1. **Efficient Memory Usage and Balanced Workload Distribution**: DEFT's KV-Guided Grouping minimizes redundant memory access by loading shared prefix data only once, reducing IO costs associated with repeatedly reloading the KV cache. Combined with the Flattened Tree KV Splitting strategy, which evenly distributes data across GPU units, DEFT maximizes GPU utilization by ensuring balanced workload distribution, thus avoiding bottlenecks and maintaining consistent processing speeds.
2. **Enhanced End-to-End Processing Speed**: Compared to state-of-the-art methods, DEFT achieves up to a 2.5x speedup in end-to-end latency, making it highly effective for tasks that require complex, tree-based structures like few-shot prompting and multi-step reasoning.
3. **Scalability Across Tasks**: DEFT demonstrates versatility by performing well across different tree-structured applications, such as speculative decoding, where shared prefix usage and efficient load balancing are particularly challenging.

**Weaknesses:**

1. **Lack of Comparison with Shared Prefix Infrastructure**: While DEFT introduces novel techniques for memory efficiency and load balancing, it lacks a direct comparison with existing infrastructure solutions like vLLM and DeepSpeed-MII, which already support shared prefix KV cache across different batches. Such a comparison would clarify DEFT’s advantages and limitations relative to widely adopted methods that also aim to reduce redundancy in KV cache management.
2. **Challenges with Distributed Memory and Tensor Parallelism**: DEFT’s current design primarily targets single-device GPU optimization and may not be directly compatible with distributed memory or tensor parallelism setups, which are commonly used to scale large language models across multiple GPUs. Adapting DEFT to work efficiently in distributed environments could require additional modifications to handle inter-device communication and memory sharing effectively, potentially limiting its scalability for very large models.

**Questions:**

1. Reasoning has become a popular approach to enhance the performance of large language models (LLMs) on complex tasks. Are there any future plans to integrate this method within task pipelines to achieve end-to-end improvements?
2. As noted in the weaknesses, tensor parallelism is widely used to scale large LLMs across multiple GPUs. Will this work be released as an open-source repository to help develop an infrastructure, similar to vLLM or DeepSpeed, that provides a usable framework for the public?
3. The test on speculative decoding sets T from 32 to 256, which is much larger than usual settings (<10), have you test speculative decoding with smaller T value?

---

> ### Author Response · Authors · 2024-11-22
> **Response to Reviewer 2d42**
>
> Thank you for your helpful feedback and positive recognition of our work.
>
> **Weaknesses:**
>
> > W1. **Lack of Comparison with Shared Prefix Infrastructure**: While DEFT introduces novel techniques for memory efficiency and load balancing, it lacks a direct comparison with existing infrastructure solutions like vLLM and DeepSpeed-MII, which already support shared prefix KV cache across different batches. Such a comparison would clarify DEFT’s advantages and limitations relative to widely adopted methods that also aim to reduce redundancy in KV cache management.
> >
>
> See  CQ2.
>
> > W2. **Challenges with Distributed Memory and Tensor Parallelism**: DEFT’s current design primarily targets single-device GPU optimization and may not be directly compatible with distributed memory or tensor parallelism setups, which are commonly used to scale large language models across multiple GPUs. Adapting DEFT to work efficiently in distributed environments could require additional modifications to handle inter-device communication and memory sharing effectively, potentially limiting its scalability for very large models.
> >
>
> See  CQ1.
>
> **Questions:**
>
> > Q1. Reasoning has become a popular approach to enhance the performance of large language models (LLMs) on complex tasks. Are there any future plans to integrate this method within task pipelines to achieve end-to-end improvements?
> >
> - We agree that reasoning is important for the future LLM serving system. Our plan consists of two parts:
>     - (1) to support more frameworks:  integrate DeFT(developed based on an early version of SGLang) to the vLLM;
>     - (2) to support reasoning frameworks: LLM Reasoners is a great framework specialized for reasoning to facilitate the development and evaluation of reasoning algorithms for LLMs.  We plan to contact them for potential cooperation to improve the efficiency of the whole reasoning pipeline, including the efficiency of attention optimized by DeFT Attention Kernel and other components like tree search.
>
> > Q2. As noted in the weaknesses, tensor parallelism is widely used to scale large LLMs across multiple GPUs. Will this work be released as an open-source repository to help develop an infrastructure, similar to vLLM or DeepSpeed, that provides a usable framework for the public?
> >
>
> Thank you for your expectations for our future work!
>
> - Tensor parallelism (TP) is completely orthogonal to the current single GPU version of DeFT, as discussed in CQ1.
> - Supporting more frameworks is definitely in our roadmap: DeFT develops based on the early version of [SGLang](https://lmsys.org/blog/2024-07-25-sglang-llama3/), an LLM serving framework that can outperform vLLM in most of the tasks. What’s more, a faster CUDA kernel is working in progress as well: the current version of DeFT attention kernel is based on Triton but a CUDA version would be faster.
> - As for why we set  SGLang as our major baseline not vLLM right now, see CQ2.
>
> > Q3. The test on speculative decoding sets T from 32 to 256, which is much larger than usual settings (<10), have you test speculative decoding with smaller T value?
> >
> - The setting of decoding tree sizes (32-256) is from Medusa, where the tree size of 64 tokens is the best in speedup discussed in the ablation study of Medusa paper: it shows a better acceleration rate than a tree size of 256 tokens with nearly the same acceptance rate of tokens, but much higher acceptance rate than small token tree sizes (e.g., 16 tokens).
> - We provided the performance when T is small (< = 10) in A100 with Llama3-8B model as follows.
>     - Setting: The prompt length is about 1k tokens.  We test Llama3-8B model in A100 80GB. The number of generated tokens is about 5K.
>     - Conclusion: We can see the speedup attention latency is still obvious(1.73-2.20x) but the end-to-end latency is 1.18-1.21x with T=5/10. The reason is that when the total number of tokens in a decoding tree is small,  the bottleneck is the FFN rather than attention.
>     - As shown in Table 18 of Appendix A7, our ablation study shows more tokens in the decoding tree( one way is to increase the prompt length), and more Attention/FFN latency ratio (A/F-LR). For long-context scenarios, attention dominates the end-to-end latency, which brings a great speedup potential on wall-clock time.
>
> | token tree size T | method | end-to-end latency (s) | attention latency (s) | attention speedup | e2e speedup |
> | --- | --- | --- | --- | --- | --- |
> | 5 | DeFT-Flatten | 44.34 | 4.15  | 1.73X | 1.18X |
> | 5 | Radix Attention | 52.38 | 7.19 | - | - |
> | 10 | DeFT-Flatten | 45.55  | 4.65 | 2.20X | 1.21X |
> | 10 | Radix Attention | 55.44  | 10.25  | - | - |

---

> > ### Comment · Reviewer_2d42 · 2024-11-26
> > **Response to Rebuttal**
> >
> > I have reviewed the author's rebuttal and appreciate their responses. I believe DEFT is a valuable contribution, and I will maintain my current rating. Best wishes to the authors, and I look forward to seeing future developments on DEFT.

---

> > > ### Author Response · Authors · 2024-11-27
> > >
> > > Thank you for taking the time to provide such constructive feedback and for recommending acceptance. Your insights and support mean a great deal to us.

---

### Official Review · Reviewer_pNpD · 2024-11-03

**Soundness:** 3
**Presentation:** 2
**Contribution:** 3
**Rating:** 8
**Confidence:** 4

**Summary:**

This paper introduces the DEFT (Decoding with Flash Tree-Attention) algorithm, aimed at enhancing efficiency in tree-structured language model (LLM) inference. Traditional approaches often fall short due to redundant memory access, inefficient handling of KV cache for shared prefixes, and poor GPU utilization. DEFT addresses these issues through two primary innovations: KV-Guided Grouping and Flattened Tree KV Splitting. Authors claim that these strategies optimize memory accesses and ensure balanced GPU utilization, leading to significant speed-ups in tree-based tasks like few-shot prompting, multi-step reasoning, and speculative decoding.

**Strengths:**

1. Authors introduce KV-Guided Grouping, which reuses memory for shared prefixes in the KV cache, minimizing redundant I/O operations.
2. Authors' approach to balanced workload distribution via Flattened Tree KV Splitting leads to better GPU usage.
3. Triton implementation provides strong empirical evidence of the efficacy of the method.

**Weaknesses:**

1. While single GPU performance is quite good, it is not clear how DeFT can scale to larger models requiring multiple GPUs.
2. Though there is a one-liner on vLLM comparison, there is no numerical comparison with vLLM given that vLLM also implements prefix-based KV-cache sharing.
3. The overhead of QKV PREPARATION PHASE is unclear from the empirical results.

**Questions:**

Please see weaknesses.

---

> ### Author Response · Authors · 2024-11-22
> **Response to Reviewer pNpD**
>
> Thank you for your helpful feedback and positive recognition of our work.
>
> **Weaknesses:**
>
> > W1. While single GPU performance is quite good, it is not clear how DeFT can scale to larger models requiring multiple GPUs.
> >
>
> See CQ1.
>
> > W2. Though there is a one-liner on vLLM comparison, there is no numerical comparison with vLLM given that vLLM also implements prefix-based KV-cache sharing.
> >
>
> See CQ2.
>
> > W3. The overhead of QKV PREPARATION PHASE is unclear from the empirical results.
> >
>
> The cost of QKV preparation phase is low— it only accounts for less than 5% of the e2e latency, while attention computation accounts for 35-70% of e2e latency.   The cost of QKV preparation is from the materialization of memory addresses of tokens for Triton—we need to serialize the memory addresses into a tensor as an input for the Triton kernel.

---

> > ### Comment · Reviewer_pNpD · 2024-12-02
> >
> > I have reviewed the author's response and appreciate their detailed answers. I believe DEFT is a significant contribution, and I will keep my current rating. All the best to the authors.

---

> > > ### Author Response · Authors · 2024-12-03
> > >
> > > We truly appreciate your support and kind words. Thanks again for taking the time to review our work and for your thoughtful feedback and encouragement.

---

### Official Review · Reviewer_M6M5 · 2024-11-09

**Soundness:** 3
**Presentation:** 2
**Contribution:** 3
**Rating:** 6
**Confidence:** 3

**Summary:**

Tree-structured decoding is gaining more popularity in LLM serving due to the presence of applications such as multi-step reasoning and speculative decoding. Existing inference systems are inefficient due to their failure to be prefix-aware: they either perform redundant recomputation of KV caches for shared prompts, or repeatedly load and store KV caches of shared prompts during attention calculation. This paper presents DeFT, an efficient attention calculation algorithm with prefix-awareness and load-balanced KV cache partitions. DeFT uses KV-guided grouping to group the prefix's KV cache with all shared queries. It then uses flattened tree KV splitting which splits the KV cache into balanced partitions to reduce overhead in computation. Evaluations show that DeFT has better wall-clock time speedup in multiple tree-structured decoding applications compared to state-of-the-art baselines.

**Strengths:**

1. Tries to solve the important problem that current LLM serving systems are inefficient in computation and IO for tree-based decoding applications.
2. Provides good background on segmented attention and existing attention algorithms.
3. Evaluation results show decent speedup over baselines.

**Weaknesses:**

1. The paper is hard to follow. The design figures include too many details. Lack of clear explanation of the key techniques including KV-guided grouping and tree KV splitting.
2. Lack of evaluation or discussion on multi-node settings and other baselines.

**Questions:**

Thank you for submitting the paper to ICLR 2025! I think this paper tries to tackle the important problem of improving GPU utilization for LLM serving under the scenario of tree-structured generation. The paper provides a good background of example tree-structured applications, how existing attention algorithms work and how attention could be calculated in a segmented way. The evaluation of the new proposed algorithm demonstrates solid speedup over existing baselines. I overall feel positive about the paper with a few comments and suggestions for improvements.

The current illustration of the main algorithm in Section 3 is hard to follow.

There are remarks and comparisons here and there.

Figure 3 includes too many notations and details that make the reader hard to recognize which are the baselines and which are the new techniques proposed in the paper. Even after reading all the text, I could not clearly figure out how flattened tree KV splitting works in detail. There are tons of places where the descriptions refer to the Appendix.
However, I think the reader should be able to grasp the intuition and how the algorithm works at a high level by just reading the main text of the paper.

My current understanding is that the core of the DeFT algorithm is to help create balanced and sharable QKV groups during the QKV Preparation Phase. It is probably better to clearly define how KV-guided grouping and flattened tree KV splitting work into two separate subsections, as they are the two main techniques proposed in the paper.

In terms of questions, how do you define the node sizes in the tree KV?

If the DeFT-Node-Chunk adds additional overhead due to imperfect splits after splitting by the nodes, could we first optimize the tree KV structure to ensure we have nodes of balanced sizes?

In the Attention Calculation Phase, how many techniques introduced in the paper are novel compared to previous works?

In addition, how does the proposed technique compare to [cascade inference algorithm](https://flashinfer.ai/2024/02/02/cascade-inference.html)? The cascade inference algorithm also makes the observation that the KV caches could be shared when there are common prefixes between requests. It first uses a multi-query attention kernel to compute the attention between queries and KV caches of the shared prefix, which goes through L1 cache and registers. Then it uses batch decode attention kernel to calculate for the remaining suffixes, which accesses the global memory and L2 cache.

In terms of experiments, it seems all evaluations are currently completed on a single A100 GPU.
How would the performance be if the algorithm is applied in a multi-node distributed LLM inference setting?
Would any of the parallelization techniques affect the effectiveness of the splitting algorithm?
How would the algorithm perform in a long context LLM serving scenario?

Other questions:

1. For Table 5, why is there an even larger speedup for the case of upper-bound (no attention)?  Isn't the proposed algorithm only optimizing for the attention operation?

2. How would different types of attention operation (e.g. multi-head, multi-query, or group-query attention) affect the performance of DeFT?

3. For Figure 4, what would the latency breakdown be for DeFT-Node-Chunk? Would unpaged versions of DeFT-Node-Chunk and DeFT-Flatten incur similar overhead for KV management?

---

> ### Author Response · Authors · 2024-11-22
> **Response to Reviewer M6M5(1/3)**
>
> **Thank you for your constructive comments and insightful questions.**
>
> **Weaknesses:**
>
> > W1. The paper is hard to follow. The design figures include too many details. Lack of clear explanation of the key techniques including KV-guided grouping and tree KV splitting.
> >
>
> See [CQ3](https://openreview.net/forum?id=2c7pfOqu9k&noteId=wDzw8mPII3) about how will we make our paper writing better.  We appreciate your feedback. We would like to point out that we already included the explanation of our two key techniques: (1) KV-guided Grouping (lines 278-285 on the left part of Figure 3, and lines 318-321); (2) Flattened Tree KV Splitting (lines 286-296 on the left part of Figure 3, and lines 351-362 );
>
> > W2. Lack of evaluation or discussion on multi-node settings and other baselines.
> >
>
> See [CQ1](https://openreview.net/forum?id=2c7pfOqu9k&noteId=HMzOOJJjzF) for a discussion on multi-node settings and CQ2 for a discussion/comparison with vLLM.
>
> > Q1: how do you define the node sizes in the tree KV?
> >
>
> **We define the node size of the tree KV by merging tokens as many as possible into a single node while avoiding the introduction of a causal mask.**
>
> - When using a traditional prefix/trie tree structure to manage tokens, each node typically represents a single token. However, this approach is not efficient for attention computation if we treat each token’s KV as a separate group when they share the same queries.
> - In essence, each node should contain as many tokens as possible, with all tokens within the node being associated with the same set of queries.
> - For example, consider a scenario where query q1 requires tokens [t1, t2, t3, t4, t5, t6] and query q2 requires tokens [t1, t2, t3, t4, t5', t6']. Here, q1 and q2 can share up to 4 tokens, [t1, t2, t3, t4], which can be grouped into a single node. If we were to merge tokens [t1, t2, t3, t4, t5, t5'] into one node, we would need a causal mask because the KV cache of t5 and t5' is only required by q1 and q2, respectively.
>
> > Q2: If the DeFT-Node-Chunk adds additional overhead due to imperfect splits after splitting by the nodes, could we first optimize the **tree KV structure to ensure we have nodes of balanced sizes?**
> >
> - First, we want to clarify that for unpaged memory, KV cache tensors are structured in a tree psychically,  while in paged memory we don’t need to do so—we just need to store the KV cache of tokens discretely in a memory pool, with records in a tree structure that maps between each token and its memory address.
>     - Therefore,  for paged memory management, we don’t need to optimize the tree KV structure in **memory storage**.
>     - Instead, we just need to optimize the logical grouping of KV cache and queries during QKV Preparation phase, for low **memory access** and great **load-balancing** in  Attention Calculation phase. In this phase, DeFT-Flatten can achieve balanced sizes of KV blocks for different QKV partitions.
> - Second, we want to point out that the cost of QKV preparation phase is low— it only accounts for less than 5% of the e2e latency, while attention computation accounts for 35-70% of e2e latency.
>     - The cost is from the materialization of memory addresses of tokens’ KV cache into a tensor as input for Triton kernel.
>     - Therefore, we don’t need to care about the cost of grouping for balanced nodes/chunks of KV cache too much.
>
> > Q3: In the Attention Calculation Phase, how many techniques introduced in the paper are novel compared to previous works?
> >
> - One of our important contributions is the insight that there is a large design space of QKV Preparation phase for tree-structured LLM inference efficiency, which is ignored by systems for sequence-based decoding.  DeFT’s main contribution lies in this phase.
> - As for Attention Calculation Phase, the existing works like Flash-Attention/Flash-Decoding are already well-designed.  The goal of this phase is to fit the QKV partitions from the previous phase as a part of the kernel/system design. The contribution of DeFT in this phase lies in the kernel design and implementation.  The global reduction of partial attention in Flash-Decoding is for sequence-based decoding, which cannot be aware of the tree-topology for global reduction in tree-based decoding. Therefore, we propose Tree-Topology-Aware Global Reduction and implement a fused kernel, as shown in Table 10 and Figure 10 b), Appendix A4.

---

> ### Author Response · Authors · 2024-11-22
> **Response to Reviewer M6M5(2/3)**
>
> > Q4: In addition, how does the proposed technique compare to [cascade inference algorithm](https://flashinfer.ai/2024/02/02/cascade-inference.html)? The cascade inference algorithm also makes the observation that the KV caches could be shared when there are common prefixes between requests. It first uses a multi-query attention kernel to compute the attention between queries and KV caches of the shared prefix, which goes through L1 cache and registers. Then it uses batch decode attention kernel to calculate for the remaining suffixes, which accesses the global memory and L2 cache.
> >
>
> Cascaded inference is one of our concurrent works. The algorithm is the same as Hygragen which we discussed in Table 9, Appendix 3.  We both have the insight that IO sharing of prefix KV cache matters. However, we have the following differences.
>
> - **Different scenarios:**
>     - Cascaded inference targets for single-context batch sampling, which only contains 2 cascades-a prefix and suffixes, as a special case of tree-based decoding;
>     - DeFT targets for general tree-based decoding with multiple-cascades, including few-shot prompting, multi-step reasoning, speculative decoding, and etc.
> - **Different challenges:**  as we target different scenarios,  we notice a trade-off between redundant calculation and load-balancing for tree-based decoding, as the node length varies a lot.
>     - If we adopt DeFT-Node/DeFT-Node-Chunk as the attention algorithm, there is no redundant calculation introduced as there is no result that would be masked, but with unbalanced workloads;
>     - If we adopt DeFT-Flatten as the attention algorithm, the partitions are balanced but invalid calculation is introduced along with causal masks.
>     - Cascaded inference does not address the challenges above.
> - **Different designs:**  We have different QKV partition strategies during QKV preparation phase.
>     - Cascaded inference groups a prefix and suffixes into two QKV groups, then combines 2 kernels for the prefix and suffixes. It works in the 2-level scenario, but in the case of multi-level cascade inference, cascaded inference doesn’t have an effective way to handle all intermediate levels.
>     - DeFT can automatically fit multiple levels of prefixes with the sharing of IOs, providing greater acceleration potential.
> - **Different implementations:**
>     - Cascaded inference cannot be expanded to multi-cascades in a single kernel.  It needs to iteratively call many kernels by users, which can introduce potential great kernel launching costs.
>     - DeFT can automatically handle multi-cascaded prefixes sharing within a single kernel, which is unified in algorithm logic and efficient in hardware.
>
> > Q5: In terms of experiments, it seems all evaluations are currently completed on a single A100 GPU. **How would the performance be if the algorithm is applied in a multi-node distributed LLM inference setting? Would any of the parallelization techniques affect the effectiveness of the splitting algorithm?** How would the algorithm perform in a long context LLM serving scenario?
> >
> - **(single-GPU for low latency, then multi-GPU for high throughput)**  The real-world serving asks us to satisfy the latency within a threshold and then improve the throughput as much as possible.  We argue that latency is the first thing we need to satisfy and it’s non-trivial already. The throughput in multiple GPUs would be the next step to be optimized in the follow-up work.  See CQ1 for details.
> - (**parallelization techniques and their impact)** Tensor parallelism is orthogonal to DeFT**.** See CQ1 for details.
> - As for a long-context LLM serving scenario, especially for the case when the prefixes are long, DeFT can even have a more obvious speedup, as shown in **Table 7.** Intuitively, this is because, for a long-context scenario, the attention computation takes most of the end-to-end latency, bringing a larger potential for wall-clock time speedup.
>
> > Q6: For Table 5, why is there an even larger speedup for the case of upper-bound (no attention)? Isn't the proposed algorithm only optimizing for the attention operation?
> >
>
> As we mentioned in the caption of **Table 5**,  Upper-bound (no attention) refers to the maximum speedup we could achieve for the best wall-clock latency baseline (Radix Attention) if we exclude the attention computation (i.e, attention latency is reduced to 0).  As we cannot reduce the attention overhead to 0, it’s the speedup upper bound of e2e latency.

---

> ### Author Response · Authors · 2024-11-22
> **Response to Reviewer M6M5(3/3)**
>
> > Q8: For Figure 4, what would the latency breakdown be for DeFT-Node-Chunk? Would unpaged versions of DeFT-Node-Chunk and DeFT-Flatten incur similar overhead for KV management?
> >
>
> The breakdown of DeFT-Node-Chunk (paged) is similar to DeFT-Flatten(paged)  but with more attention overhead:
>
> | Method | End to End (s) | Attention Computation (s, % of End to End) | KV Management (s, % of End to End) |
> | --- | --- | --- | --- |
> | DeFT-Flatten | 49.01 | 14.28 (29.13%) | 6.76 (13.79%) |
> | DeFT-Node | 89.19 | 51.97 (58.26%) | 6.84 (7.67%) |
> | DeFT-Node-Chunk | 53.44 | 16.48 (30.84%) | 6.68 (12.51%) |
> | Radix Attention | 69.96 | 35.71 (51.05%) | 6.73 (9.62%) |
>
> *Table: Latency breakdown for speculative decoding with a token tree of 32 queries, whose tree topology is from Medusa, in seconds. Values in parentheses represent the percentage of the end-to-end time.*
>
> - We don’t implement unpaged versions of DeFT-Node-Chunk and DeFT-Flatten, because the comparison between DeFT-Node (paged) and DeFT-Node (unpaged) can show the superiority of paged memory management already.
> - Theoretically, DeFT-Node (unpaged) and DeFT-Node-Chunk (unpaged) should incur similar overhead for KV management with DeFT-Node (unpaged), where the materialization/concatenation of KV cache from a tree structure to a single tensor for attention calculation is expensive.
> - As for paged memory management, DeFT-Node, DeFT-Node-Chunk, and DeFT-Flatten should have nearly the same memory management cost as the difference between these three lies in the memory access of KV cache, not the memory storage.

---

### Author Response · Authors · 2024-11-22
**Common Response(1/3)**

We appreciate the reviewers for their insightful comments and constructive feedback. We are pleased to note that all reviews were positive and that the reviewers recognized our work as addressing a significant problem.

We would like to first address some common questions, and then respond to specific inquiries raised by the reviewers. The improvements to the manuscript have been finished and the modified part is in **ORANGE**, please refer to Common Question 3 (CQ3) for an outline of updates.

## **Questions in common(CQs)**

### CQ1

> CQ1: can DeFT be extended to multi-GPU versions? What’s the performance of DeFT in multiple GPUs setting?
>
- We would like to point out that the single-GPU version of DeFT targets low latency (request finish time), which is non-trivial already, as there are tradeoffs between redundant memory access/calculation and load-balancing. The LLM serving asks us to reduce the latency to below a threshold and then improve the throughput as much as possible. Improving the throughput in multiple GPUs without sacrificing the latency too much would be the future step, which could be achieved by a better design of batching and scheduling.
- Regarding the **parallelization techniques and their impact**
    - Tensor parallelism(TP) is completely orthogonal to the current single-GPU version of DeFT because TP does partitioning in the head for attention and hidden dimension for MLP, while DeFT does partitioning in the dimension of sequence length(s). TP introduces two allreduce communication, in the attention module, the head dimension is split to distribute the computation over different GPUs, in this case we just need to manage the KV of each head separately on multiple GPUs, in the MLP module, the intermediate hidden dimension is partitioned, which does not modify any implementation of our approach.
    - If we want to have sequence parallelism(SP) in multiple GPUs, it’s non-trivial in the system design because there could be problems like KV cache fragmentation and the trade-off between communication of partial attention and KV cache movement for better locality. There are some latest works[1][2] about sequence parallelism for general sequence-based serving systems that explored this topic.  Tree-based decoding will make it more challenging, but we are interested in exploring this topic in the future.

*[1] Sun, B., Huang, Z., Zhao, H., Xiao, W., Zhang, X., Li, Y., & Lin, W. (2024). Llumnix: Dynamic Scheduling for Large Language Model Serving. In Proceedings of the 18th USENIX Symposium on Operating Systems Design and Implementation (OSDI 24).*

*[2] B. Wu, S. Liu, Y. Zhong, P. Sun, X. Liu, and X. Jin, “Loongserve: Efficiently serving long-context large language models with elastic sequence parallelism,”* In Proceedings of the ACM SIGOPS 30th Symposium on Operating Systems Principles (SOSP '24).

---

> ### Author Response · Authors · 2024-11-22
> **Common Response(2/3)**
>
> ### CQ2
>
> > CQ2: the comparison between DeFT and vLLM with its prefix-caching?
>
> We argue that the direct comparison with vLLM is unfair and prefix-caching is a technique for a faster prefill stage, which is not the bottleneck and orthogonal to DeFT’s optimization on the decoding stage.
>
> - First, we want to point out that prefix-caching is a technique of memory storage optimization with the benefit of prefill phase acceleration, not a technique that optimizes memory access for decoding phase speedup.
>     - SGLang (DeFT develops based on an early version of SGLang) adopts a radix tree and vLLM adopts hash codes to maintain the KV cache storage with prefix-awareness.
>     - Prefix-caching can only reduce the Time To First Token (*TTFT*) in the prefill phase, while it only takes a very small percentage (<5% in most of the tasks we test, as it takes <2 seconds for 4k prompt length in Llama3-8B) of end-to-end latency for tasks with long generation lengths.  It does not reduce the time needed to generate new tokens (the decoding phase).
>     - Instead, DeFT optimized time Per Output Token (*TPOT*) in the decoding phase by memory access reduction *(KV-guided grouping)* and balanced partitions *(Flattened tree splitting)* of KV cache, and the prefix-caching is just as an orthogonal technique for storage optimization in DeFT.
> - Second, as a baseline of attention kernel, we argue that SGLang is fairer and more reliable because we can control every part the same except the attention kernel for comparison of performance. We would like to clarify that the current version of DeFT was developed based on SGLang,  which is the first framework that supports flexible tree-structured management of KV cache along with an attention kernel (Radix Attention) that fits such management.  We chose and prioritized SGLang as our major baseline for three reasons:
>     1. It adopts radix tree for **Automatic KV Cache Reuse in storage level(DeFT can even reuse KV cache in memory access level)**, which is more flexible than vLLM;
>     2. Based on the [SGLang’s documentation](https://lmsys.org/blog/2024-07-25-sglang-llama3/), **SGLang can even outperform vLLM in many workloads;**
>     3. Besides, Radix attention kernel in SGLang is developed based on Triton (DeFT attention kernel is also on Triton), while attention kernels in vLLM are based on CUDA.   Actually, the attention algorithm in these two frameworks is the same and the major difference is the implementation in Triton/CUDA.  Therefore, Radix Attention is a fairer baseline of DeFT that can reduce the difference in implementation to show the effectiveness of DeFT algorithms.
> - Third,  although it is not fair to compare with vLLM’s attention kernel because CUDA’s implementation has a natural speed advantage over Triton (e.g, Flash-Attention on Triton would be 70% speed of Flash-Attention on CUDA), we still provide the comparison results with DeFT to prove the effectiveness of the DeFT-Flatten algorithm.
>     - [Experiment setting] Radix Attention and DeFT-Flatten is based on SGLang. The workload is few-shot prompting tokens in A100 80GB with tree width=1/5/20/30. The model is Llama3-8B. The prompt length is about 1k tokens.
>     - [Notions] The best latency is in **bold**, and the second best is in *italic*.
>     - **[Conclusion]** When the treewidth is small (<20), the attention latency of Paged Attention in vLLM is the best as it is based on CUDA while Radix Attention and DeFT-Flatten are based on Triton.  When the treewidth is 30, DeFT-Flatten is faster than Paged Attention in attention/end-to-end latency with 1.25X/1.20X speedup, respectively. This is because the advantages of the DeFT-Flatten algorithm in memory access overcome the disadvantages of the implementation (Triton V.S. CUDA).
>
> | treewidth | framework+Attention kernel | end-to-end latency | attention latency |
> | --- | --- | --- | --- |
> | 1 | SGLang+Radix Attention | 10.64 | 3.82 |
> | 1 | vLLM+Paged Attention | **8.37** | **2.01** |
> | 1 | SGLang+DeFT-Flatten | *9.32* | *2.43* |
> | 5 | SGLang+Radix Attention | 11.07 | 4.21 |
> | 5 | vLLM+Paged Attention | **9.25** | **2.43** |
> | 5 | SGLang+DeFT-Flatten | *9.57* | *2.67* |
> | 20 | SGLang+Radix Attention | *12.37* | 5.99 |
> | 20 | vLLM+Paged Attention | 12.58 | **4.19** |
> | 20 | SGLang+DeFT-Flatten | **11.82** | *4.20* |
> | 30 | SGLang+Radix Attention | *14.08* | 7.30 |
> | 30 | vLLM+Paged Attention | 15.18 | *6.18* |
> | 30 | SGLang+DeFT-Flatten | **12.69** | **4.94** |

---

> ### Author Response · Authors · 2024-11-22
> **Common Response(3/3)**
>
> ### CQ3
>
> > CQ3: how to improve the paper manuscript?
> >
>
> We thank all reviewers for their detailed reviews. We have made the following changes (in **ORANGE** ) to the paper and uploaded a new revision:
>
> - (suggested by reviewer **M6M5**) reorganize the elements in  Figure 3 to distinguish our main techniques and baselines. We separated the baselines (Flash-Decoding, Radix-Attention, etc) and main techniques of DeFT (KV-guided Grouping and Flattened Tree KV Splitting) into three subgraphs.
> - (suggested by reviewer **M6M5)** modify Figure 4 by adding the latency breakdown of DeFT-Node-Chunk .
> - (suggested by reviewer **M6M5 and 6baS**) reorganized and reduced the call to the appendix in the main text of section 3, to make sure the reader can get the key design of DeFT intuitively and find details in the Appendix.

---

### Author Response · Authors · 2024-11-25
**Appreicate the efforts of all reviewers**

We thank all reviewers for their constructive and insightful efforts in evaluating this work. We have uploaded revised files, with several modifications (in **ORANGE**) :
- (suggested by reviewer **M6M5**) reorganize the elements in  Figure 3 to distinguish our main techniques and baselines. We separated the baselines (Flash-Decoding, Radix-Attention, etc) and main techniques of DeFT (KV-guided Grouping and Flattened Tree KV Splitting) into three sub-figures.
- (suggested by reviewer **M6M5)** modify Figure 4 by adding the latency breakdown of DeFT-Node-Chunk .
- (suggested by reviewer **M6M5 and 6baS**) reorganized and reduced the call to the appendix in the main text of section 3, to make sure the reader can get the key design of DeFT intuitively and find details in the Appendix.

We greatly appreciate the time and effort you've dedicated to reviewing our paper. Your feedback has played a crucial role in enhancing the quality of our manuscript.  If you have any additional questions or comments, please feel free to reach out.

Authors

---

### Meta-Review · Area_Chair_hQU3 · 2024-12-17

**Metareview:**

This paper presents DeFT, a novel algorithm for enhancing tree-based decoding in LLM inference. It targets the inefficiencies of existing systems, such as redundant KV cache access and poor load balancing. By introducing techniques like KV-Guided Grouping and Flattened Tree KV Splitting, DeFT aims to optimize memory access and workload distribution. Empirically, it achieves significant speedup in end-to-end and attention latency compared to current state-of-the-art methods.

The paper's strengths lie in its timeliness and relevance, with a novel and theoretically sound approach supported by solid empirical evidence. However, it has weaknesses. The main text could be clearer, with some crucial details in the supplementary material. There's a lack of energy efficiency analysis, and further exploration in specific scenarios is needed.

**Additional Comments On Reviewer Discussion:**

During the rebuttal period, reviewers raised points regarding clarity of presentation (suggesting integrating supplementary material into the main text), energy efficiency measurement, performance in scenarios with minimal shared prefixes or small tree widths, and realistic scalability to larger model sizes. The authors addressed these by outlining changes made to improve paper clarity, explaining the plan to explore energy efficiency in future work, presenting performance data in relevant scenarios, and discussing challenges and possible solutions for extending DeFT to larger models. These responses showed the authors' thorough consideration of the issues and their commitment to improving the work, which weighed positively in the final decision.

---

### Decision · Program_Chairs · 2025-01-22

Accept (Spotlight)